# Blocking glycine utilization inhibits multiple myeloma progression by disrupting glutathione balance

Jiliang Xia[1,2,3,11], Jingyu Zhang[1,2,11], Xuan Wu[2], Wanqing Du[4], Yinghong Zhu[1,2], Xing Liu[1,2], Zhenhao Liu [1,2], Bin Meng[1,2], Jiaojiao Guo[2], Qin Yang[5], Yihui Wang[1,2], Qinglin Wang[6], Xiangling Feng[4], Guoxiang Xie [7], Yi Shen[8], Yanjuan He[1], Juanjuan Xiang [2], Minghua Wu [2], Gang An[9], Lugui Qiu [9], Wei Jia [10] & Wen Zhou [1,2✉]

Metabolites in the tumor microenvironment are a critical factor for tumor progression. However, the lack of knowledge about the metabolic profile in the bone marrow (BM) microenvironment of multiple myeloma (MM) limits our understanding of MM progression. Here, we show that the glycine concentration in the BM microenvironment is elevated due to bone collagen degradation mediated by MM cell-secreted matrix metallopeptidase 13 (MMP13), while the elevated glycine level is linked to MM progression. MM cells utilize the channel protein solute carrier family 6 member 9 (SLC6A9) to absorb extrinsic glycine subsequently involved in the synthesis of glutathione (GSH) and purines. Inhibiting glycine utilization via SLC6A9 knockdown or the treatment with betaine suppresses MM cell proliferation and enhances the effects of bortezomib on MM cells. Together, we identify glycine as a key metabolic regulator of MM, unveil molecular mechanisms governing MM progression, and provide a promising therapeutic strategy for MM treatment.

[1] Haihe Laboratory of Cell Ecosystem, State Key Laboratory of Experimental Hematology, Key Laboratory for Carcinogenesis and Invasion, Chinese Ministry of Education, Key Laboratory of Carcinogenesis, Chinese Ministry of Health, Department of Hematology, Xiangya Hospital, Central South University, Changsha, Hunan, China. [2] Cancer Research Institute, School of Basic Medical Sciences, Central South University, Changsha, Hunan, China. [3] Hunan Province Key Laboratory of Tumor Cellular & Molecular Pathology, Cancer Research Institute, Hengyang Medical School, University of South China, Hengyang, Hunan, China. [4] Xiangya School of Public Health, Central South University, Changsha, Hunan, China. [5] Department of Hematology, Third Xiangya Hospital, Central South University, Changsha, Hunan, China. [6] Department of Laboratory Medicine, Hunan Normal University School of Medicine, Changsha, Hunan, China. [7] Human Metabolomics Institute, Inc, Shenzhen, Guangdong, China. [8] Department of Orthopaedic Surgery, Second Xiangya Hospital, Central South University, Changsha, Hunan, China. [9] State Key Laboratory of Experimental Hematology, Institute of Hematology & Blood Diseases Hospital, Chinese Academy of Medical Science & Peking Union Medical College, Tianjin, China. [10] School of Chinese Medicine, Hong Kong Baptist University, Kowloon Tong, Hong Kong, China. [11] These authors contributed equally: Jiliang Xia, Jingyu Zhang. ✉email: wenzhou@csu.edu.cn

Multiple myeloma (MM) is characterized by transformed clonal plasma cells in the bone marrow (BM) microenvironment and monoclonal immunoglobulin accumulation in the blood or urine[1]. Recent findings from our group and others have shown that genetic variations and the BM microenvironment contribute to malignant progression and anticancer therapy resistance in MM[1,2]. Metabolic reprogramming in the tumor microenvironment reportedly promotes tumor progression in various types of cancer[3–5]. More recently, we demonstrated that intestinal bacteria promote MM progression by elevating the level of BM glutamine in MM[6], indicating that the alterations of metabolite levels in the BM microenvironment can affect MM.

The metabolic profile is strongly influenced by pathological changes. Tumor initiation results in metabolite alterations in the tumor microenvironment which, in turn, contribute to tumor progression[7–10]. Recently, it has been shown that tumor-associated metabolites can potentially be used for clinical diagnosis and therapy in cancer. For instance, Tan et al. found that a panel of five plasma metabolites can serve as a noninvasive diagnostic marker for pancreatic cancer[11], and Chen et al. reported that blocking fructose utilization attenuates proliferation and drug resistance in acute myeloid leukemia[12]. However, a detailed understanding of the metabolite profile in the MM BM microenvironment is still lacking. The function of metabolites in MM progression is even more poorly understood.

In this work, we attempt to fill the gap in our knowledge of metabolites involved in MM. We perform a metabolomic analysis of BM liquid and peripheral blood (PB) plasma-derived from MM patients and healthy donors (HDs). Finally, we examine MM-induced alterations of metabolites and their therapeutic potential for the treatment of MM.

## Results

### Glycine is upregulated in the BM microenvironment and is linked to poor MM prognosis.
We performed untargeted metabolomics studies using PB plasma and BM liquid derived from HDs and MM patients. The BM liquid samples used for these analyses were divided into a training set and a validation set. To screen the differentially regulated metabolites between HDs and MM patients, we analyzed the spectral intensities of metabolites using SIMCA-P 14.1 software and MetaboAnalyst 5.0. Orthogonal partial least-squares-discriminant analysis (OPLS-DA) models were established and showed satisfactory modeling and predictive abilities for distinguishing MM patients from HDs in all datasets (Fig. 1a). The intercept of Q2Y (the predictive accuracy of the model) with a threshold less than zero indicated a valid model in all sets (Supplementary Fig. 1a). Variable importance of the projection (VIP) values of metabolites were obtained from these OPLS-DA models, and the fold changes (FC) and p-values of metabolites were obtained with MetaboAnalyst 5.0 based on the intensities of the spectrum. The detailed VIP values, FCs, and p-values of metabolites are shown in Supplementary Data 1. Most of the amino acids were upregulated in MM patients compared with HDs, while fatty acids were downregulated (Supplementary Fig. 1b). In total, 10 metabolites were identified as differential metabolites in BM liquid in both the training set and the validation set (VIP > 1, FC > 1.2, $p < 0.05$). The identified metabolites in BM liquid included glycine, L-proline, creatinine, ornithine, L-glutamic acid, L-valine, stearic acid, palmitic acid, petroselinic acid, and linoleic acid, among which seven metabolites were also significantly altered in PB plasma (Fig. 1b and Supplementary Data 2). Intriguingly, glycine showed the highest VIP values in the BM-validation set and in PB plasma, suggesting an important role in MM.

Subsequently, we confirmed that glycine concentrations in both BM liquid and PB plasma were significantly elevated in MM patients when compared with HDs (Fig. 1c). The glycine concentrations in PB plasma and BM liquid in the same MM patients were also positively correlated ($r^2 = 0.2197$, $p < 0.0001$) (Fig. 1d), implying that glycine is a potentially valuable diagnostic biomarker for MM. We subsequently evaluated this potential by producing a receiver operator characteristic curve (ROC) of glycine concentrations. ROC analyses showed that glycine in both BM liquid and PB plasma was effective in discriminating newly diagnosed MM patients from HDs, with the area under the curve (AUC) values being 1 and 0.94 respectively (Fig. 1e). To explore whether glycine played a role in MM, we assessed the potential correlation between BM glycine concentration and the clinical characteristics of MM. We found that the glycine concentration correlated strongly with the percentage of plasma cells ($p = 0.043$), the Durie-Salmon (DS) stage ($p = 0.018$), the subtype of immunoglobulin present in serum ($p = 0.04$), and anemia ($p = 0.008$) (Table 1). These findings suggested that glycine is linked to poor prognosis and malignant progression in MM patients.

### Glycine deprivation inhibits myeloma cell proliferation in vitro and in vivo.
In keeping with the metabolic analyses, human MM cell lines (including ARP1, MM1.S, RPMI-8226, OCI-My5, and KMS28-PE) all exhibited higher intracellular glycine concentrations than the GM12878, human B cell line (Supplementary Fig. 2a). Moreover, upregulation of the glycine transporter solute carrier family 6, member 9 (SLC6A9) was observed in ARP1, OCI-My5 and KMS28-PE when compared with GM12878 cells, while the elevation of another glycine transporter (SLC6A5) was not observed in all MM cell lines relative to GM12878 cells (Supplementary Fig. 2b). We thus hypothesized that MM cells absorbed more glycine from the culture media relative to GM12878 control cells.

As expected, MM cell lines cultured in the presence of glycine showed higher intracellular glycine concentrations (Fig. 2a), while deprivation of exogenous glycine significantly inhibited the proliferation of MM cells and reduced colony numbers (Fig. 2b, c). To test whether exogenous glycine deprivation also suppressed MM progression in vivo, we fed MM-prone C57BL/KalwRijHsd mice a normal or glycine-free diet for one week and then injected intravenously with luciferase-expressing 5TGM1 mouse myeloma cells. We monitored MM development and the parameters of tumor burden in these mice (dubbed "5TGM1 MM mice") via whole-animal live imaging and measurement of IgG2b concentration and bone damage (Fig. 2d). The serum glycine concentrations in control mice were much higher than those in glycine-free mice at week 6 (Fig. 2e). Compared with the control mice, the mice fed a glycine-free diet exhibited significantly reduced tumor-associated luminescence intensity at week 8 after injection of 5TGM1 cells (Fig. 2f, g). Moreover, the concentrations of serum IgG2b in the control mice were significantly higher than those in the glycine-free mice at weeks 4 and 8 (Fig. 2h).

Bone damage is one of the most important features of MM. We thus used micro-CT scanning to detect bone damage in mouse femurs, finding that the control mice had markedly lower femoral trabecular bone mass compared to the glycine-free mice (Fig. 2i). Quantification of bone microstructural parameters showed that the trabecular bone volume fraction (Tb. BV/TV) and trabecular number (Tb. N) in the control mice were much lower than those in glycine-free mice, while the trabecular separation (Tb. Sp) and trabecular plate thickness (Tb.Th) were elevated in the control mice compared with the glycine-free mice (Fig. 2j). Moreover, bone resorption marker CTX-I (C-terminal telopeptide of type I collagen) and bone formation marker PINP (Procollagen type

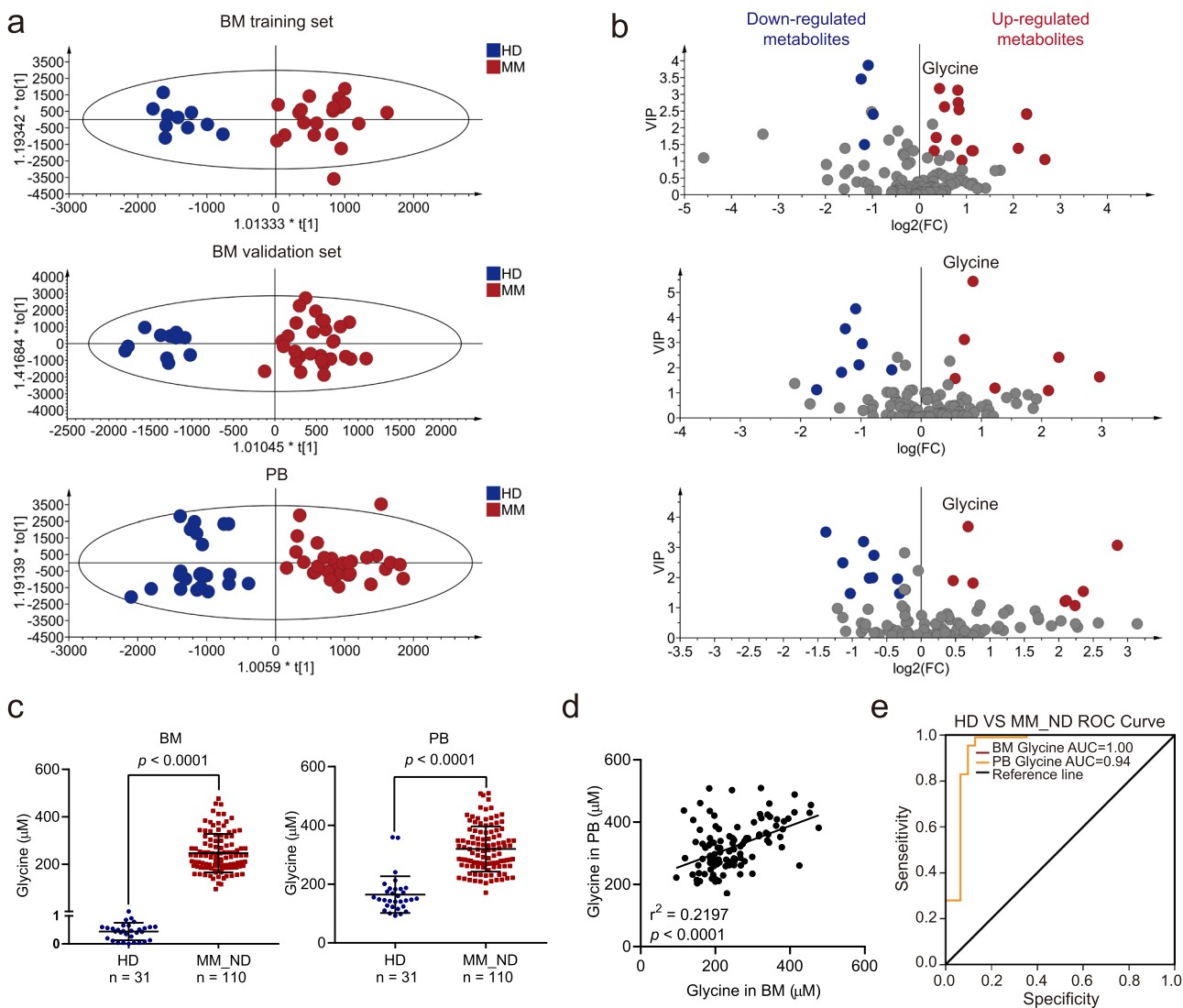

**Fig. 1 Glycine is elevated in the BM microenvironment and linked to poor prognosis in MM. a, b** (a) Plots of OPLS-DA scores for the BM liquid and PB plasma derived from MM patients and HDs (BM-training set: $n = 10$ in HD group, $n = 20$ in MM group; BM-validation set: $n = 11$ in HD group, $n = 30$ in MM group; PB: $n = 26$ in HD group, $n = 31$ in MM group), and (b) differential metabolites in all sets based on VIP, FC, and p-value (MetaboAnalyst 5.0 software with unpaired two-sided t-test). **c** Left: targeted metabolomics assays of glycine in the BM liquid derived from HDs ($n = 31$) and newly diagnosed MM patients ($n = 110$); Right: targeted metabolomics assays of glycine in the PB plasma derived from HDs ($n = 31$) and newly diagnosed MM patients ($n = 110$). Results represent means ± SD; A unpaired two-sided t-test was applied. **d** The correlation of glycine concentrations from the BM liquid and PB plasma ($n = 110$; two-sided Pearson correlation analysis). **e** ROC curves drawn based on glycine concentrations using subjects derived from HDs and newly diagnosed MM patients ($n = 110$). Source data are provided as a Source Data file.

I N-terminal propeptide), which are elevated in patients with osteoporosis[13], were examined in mcie serum. Both CTX-I and PINP were higher in the control mice compared to glycine-free mice, further revealing more serious bone damage in the control mice (Fig. 2k, l). In addition, analysis of the survival curves showed that the glycine-free diet markedly extended mouse survival when compared with the control diet (Fig. 2m). Thus, exogenous glycine deprivation slows MM progression in vivo.

**Glycine deprivation inhibits proliferation by disrupting the balance of intracellular GSH in myeloma cells.** To explore the underlying mechanisms by which exogenous glycine modulates proliferation in MM cells, we cultured ARP1 cells in glycine-free RPMI 1640 media supplemented with 5% dialyzed FBS and isotope-labeled glycine (10 mg/L). To eliminate interference caused by the natural isotope, cells collected at 0 h were used as a

reference for analysis. The cells were then processed for examination of glycine, serine, GSH, cysteine, glutamic acid, homo-cysteine, inosine-5′-monophosphate (IMP), guanosine 5-monophosphate (GMP), adenosine 5-monophosphate (AMP), adenine and guanine levels with or without $^{13}$C labeling (Fig. 3a). Metabolic tracing with $^{13}$C$_2$-glycine showed that glycine was incorporated into serine (m + 2), GSH (m + 1 and m + 2), IMP (m + 2 and m + 3), GMP (m + 2 and m + 3), AMP (m + 2 and m + 3), adenine (m + 2) and guanine (m + 2) in ARP1 cells (Supplementary Fig. 3a). Thus, exogenous glycine is absorbed by MM cells and subsequently metabolized into these metabolites by providing carbons (Fig. 3b). The mass distribution vectors (MDVs) of $^{13}$C-labeled metabolites were also found to increase over time (Supplementary Fig. 3b–d). To explore whether exo-genous glycine promotes cell proliferation via GSH or purines in MM cells, we cultured ARP1 cells with glycine-free media sup-plemented with GSH (10 μM) or hypoxanthine (50 μM), and a

**Table 1 The correlations between glycine concentration and clinical characteristics of MM patients.**

| Patients' characteristics | Low glycine (n = 27; n/N*100%) | High glycine (n = 83; n/N*100%) | p-value |
|---|---|---|---|
| *Sex* | | | 0.952 |
| Male | 14/27(51.85) | 43/83(51.81) | |
| Female | 13/27(48.15) | 40/83(48.19) | |
| *Age (Year)* | 57.07 ± 10.51 | 58.51 ± 10.59 | 0.535 |
| *Subtype of immunoglobulin* | | | 0.040 |
| IgA | 2/23(8.70) | 22/69(31.88) | |
| IgG | 20/23(86.96) | 40/69(57.97) | |
| IgD | 1/23 (4.34) | 7/69(10.15) | |
| *Subtype of immunoglobulin light chain* | | | 0.666 |
| κ | 15/27(55.56) | 42/83(50.60) | |
| λ | 12/27(44.44) | 41/83(49.40) | |
| *DS stage* | | | 0.018 |
| I | 3/27(11.11) | 1/83(1.20) | |
| II + III | 24/27(88.89) | 82/83(98.80) | |
| *ISS stage* | | | 0.496 |
| I | 7/27(25.93) | 14/82(17.07) | |
| II | 8/27(29.63) | 22/82(26.19) | |
| III | 12/27(44.44) | 46/82(53.66) | |
| *IgH translocation* | | | 0.635 |
| No | 6/24(25.00) | 24/79(30.38) | |
| Yes | 18/24(75.00) | 55/79(69.62) | |
| *Anemia* | | | 0.008 |
| No | 13/27(48.15) | 18/83(21.69) | |
| Yes | 14/27(51.85) | 65/83(77.31) | |
| LDH(U/L) | 175.5 ± 56.45 | 175.29 ± 90.29 | 0.293 |
| Plasma cells in bone marrow (%) | 35.00 ± 27.08 | 51.36 ± 28.73 | 0.043 |
| $Ca^{2+}$ (mmol/L) | 2.43 ± 0.45 | 2.29 ± 0.31 | 0.202 |
| Platelet ($\times 10^9$/L) | 166.59 ± 74.15 | 182.01 ± 84.95 | 0.59 |
| Creatinine (μmol/L) | 105.87 ± 71.29 | 119.12 ± 123.57 | 0.7 |

The correlations between glycine concentration and clinical characteristics were measured using the chi-square test. Source data are provided as a Source Data file.

colorimetric cell viability assay (Cell Counting Kit-8, CCK-8) was used three days later to measure cell proliferation. As shown in Supplementary Fig. 3e, both GSH and hypoxanthine could rescue proliferation in ARP1 cells cultured in glycine-free medium. Previous publication indicated that exogenours glycine contributed to purines synthesis rather than GSH synthesis in melanoma cells[14]. In this study, we thus focused on the role of GSH derived from exogenours glycine in MM. Furthermore, treatment with GSH increased ARP1 cell proliferation over time (Fig. 3c), and increased exogenous glycine correspondingly increased intracellular GSH levels (Fig. 3d). The same pattern was observed in 5TGM1 cells (Supplementary Fig. 3f). Increased exogenous glycine also reduced reactive oxygen species (ROS) levels in ARP1 and 5TGM1 cells (Supplementary Fig. 3g). We further found that GSH levels were significantly lower in tumors derived from glycine-free diet-fed xenograft MM mice than in those from mice fed a normal diet (Fig. 3e). Tumors developed in glycine-free mice were also significantly smaller than those in control mice (Supplementary Fig. 3h).

Inhibition of GSH synthesis reportedly suppresses tumor cell proliferation in various types of cancer[15–17]. We found that disrupting intracellular GSH balance by preventing GSH synthesis or by adding exogenous GSH inhibited the proliferation of both ARP1 and 5TGM1 cells (Supplementary Fig. 3i, j). We thus investigated whether the addition of GSH restored tumor progression in 5TGM1 MM mice fed with the glycine-free diet. Tumor burden assays showed that both tumor-associated

luminescence intensity (Fig. 3f, g) and serum IgG2b concentrations (Fig. 3h) were significantly higher in 5TGM1 MM mice treated with GSH than in the control mice. Mice treated with GSH also showed remarkably shortened survival time (Fig. 3i). Thus, GSH acts as a downstream effector of glycine to mediate MM progression.

RNA-sequencing using ARP1 cells cultured with or without glycine showed that SLC6A9, Glycine decarboxylase (GLDC), Serine hydroxymethyltransferase 2 (SHMT2), and Glutathione synthetase (GSS) were significantly upregulated in the presence of exogenous glycine (Fig. 3j and Supplementary Data 3). This suggested that exogenous glycine deprivation induces alterations in glycine transport, glycine cleavage, the interconversion between glycine and serine, and GSH synthesis, consistent with the glycine metabolic flux analysis (Fig. 3k). Analysis of enriched signaling pathways based on differentially expressed genes showed that half of the top 10 enriched pathways were related to cell mitosis and DNA damage (Supplementary Fig. 3k and Supplementary Data 4). Subsequent analyses of cell cycle and DNA damage revealed that exogenous glycine deprivation-induced S-phase arrest (Supplementary Fig. 3l) and the upregulation of γ-H$_2$AX, a marker for DNA damage (Fig. 3l and Supplementary Fig. 3m). Because ROS-induced DNA damage can activate DNA repair pathways and result in cell cycle arrest (Supplementary Fig. 3n), we performed western blotting to examine the expression of γ-H$_2$AX and DNA repair-related proteins in these cells. We found that GSH treatment reduced the levels of γ-H$_2$AX, p-ATR, and p-CHK1, while upregulating CDC25A in ARP1 cells cultured in glycine-deprived media. In contrast, the phosphorylation of ATM and CHK2 was unaffected by GSH (Fig. 3m). These findings demonstrated that exogenous glycine deprivation induces DNA damage and subsequently activates the ATR-CHK1 DNA repair pathway, leading to S-phase arrest in MM cells.

**Inhibition of GLDC suppresses proliferation in MM cells.** The glycine cleavage system (GCS) consists of several proteins, including glycine dehydrogenase (GLDC), GCS protein H (GCSH), aminomethyltransferase (AMT), and dihydrolipoamide dehydrogenase (DLD), which catalyze the following reversible reaction: Glycine + H$_4$folate + NAD$^+$ ⇌ 5,10-methylene-H$_4$folate + CO$_2$ + NH$_3$ + NADH + H$^+$ (Fig. 4a)[18]. We found that exogenous glycine deprivation reduced the ratio of NADH / NAD$^+$ in ARP1 and 5TGM1 cells (Fig. 4b), suggesting that exogenous glycine could be cleaved by the GCS system. The above RNA-sequencing data showed that GLDC was downregulated by exogenous glycine deprivation in ARP1 cells, and this was further confirmed by western blotting (Fig. 4c). To determine whether GLDC played a role in MM, we knocked down GLDC in ARP1 cells (Fig. 4d). We found that GLDC knockdown inhibited proliferation (Fig. 4e), reduced GSH levels (Fig. 4f), and decreased the ratio of NADH / NAD$^+$ (Fig. 4g) in ARP1 cells.

Next, we provided exogenous GSH to GLDC-depleted cells and found that this restored proliferation and cell colony formation in ARP1 cells after GLDC knockdown (Fig. 4h, i). Knockdown of GCSH or AMT also suppressed the proliferation of ARP1 cells, however, knockdown of DLD, which catalyzes the interconversion between NAD$^+$ and NADH, did not significantly affect ARP1 cell proliferation (Supplementary Fig. 4a–f). In addition, we found that knockdown of GLDC increased γ-H$_2$AX, p-ATR, and p-CHK1 and decreased CDC25A protein levels in ARP1 cells, while the addition of GSH reversed these alterations (Fig. 4j). Thus, GLDC promotes the production of GSH, while knockdown of GLDC induces DNA damage and consequently inhibits the proliferation of MM cells.

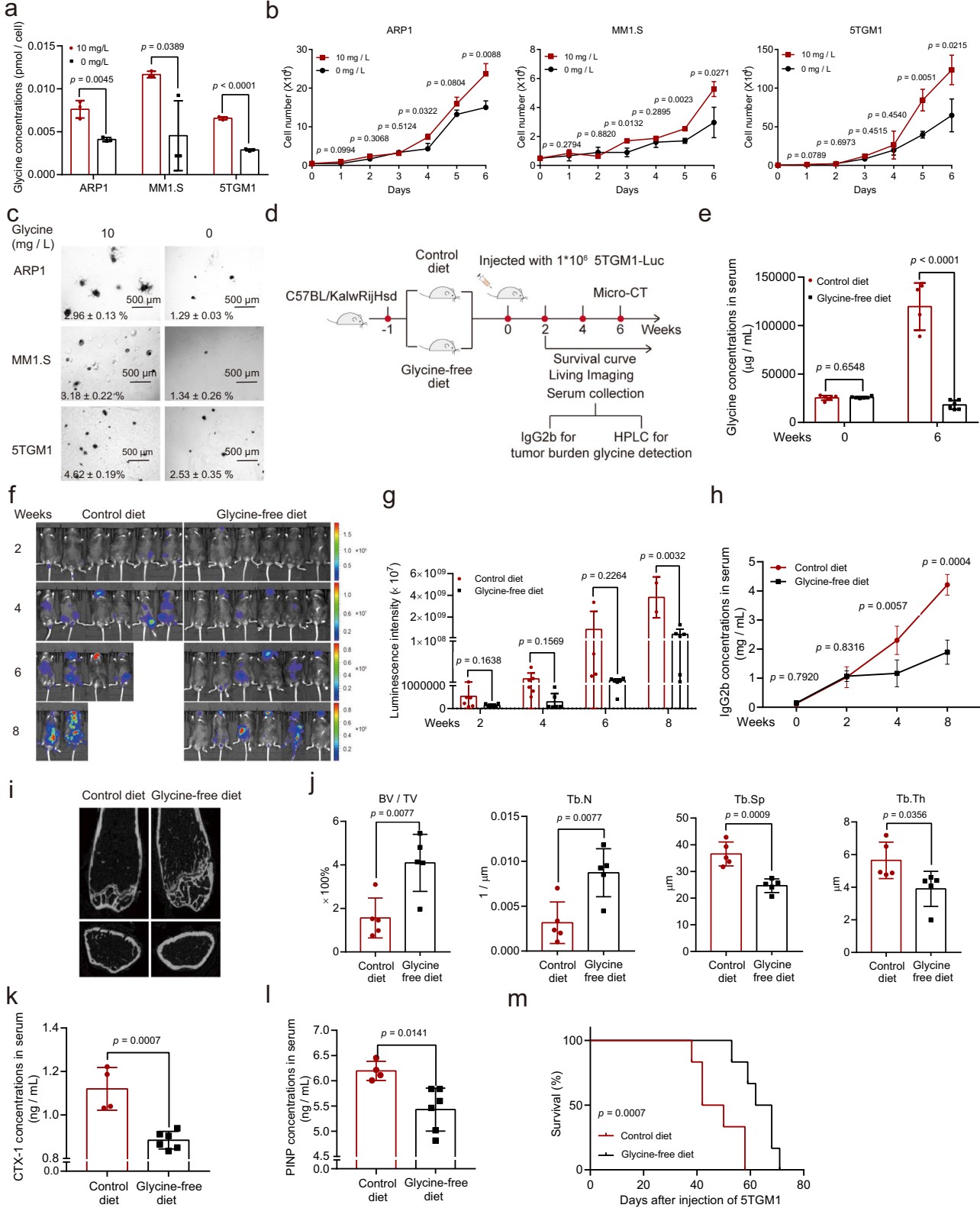

We next used immunocompromised B-NDG mice to examine the effect of GLDC knockdown in vivo. We injected B-NDG mice with ARP1 cells expressing GLDC shRNA1 or scrambled shRNA subcutaneously and induced shRNA expression by adding doxycycline (2 mg/mL) to the drinking water one week after the engraftment of tumor cells. We also supplied GSH (2 mg/kg) to half of the mice transplanted with ARP1 GLDC shRNA1 cells. As

shown in Fig. 4k, l, the GLDC-knockdown tumors were smaller than those expressing scrambled shRNA, and the addition of GSH increased the tumor size in mice with GLDC shRNA1. GSH assays showed that GLDC depletion reduced GSH levels in the tumors, while GSH addition expectedly restored the GSH levels in tumors containing GLDC shRNA1 (Fig. 4m). Consistent with the results from the in vitro experiments, GDLC knockdown elevated

**Fig. 2 Deprivation of exogenous glycine inhibits MM cell growth both in vitro and in vivo. a** Targeted assays of glycine in ARP1, MM1.S, and 5TGM1 cells cultured in RPMI 1640 media supplemented with or without glycine (10 mg/L). **b** Growth curves of ARP1, MM1.S, and 5TGM1 cells cultured in RPMI 1640 media supplemented with or without glycine (10 mg/L). $n = 3$ independent experiments; Results represent means ± SD; Significance was analyzed with an unpaired two-sided $t$-test in (**a**, **b**). **c** Clonogenic analysis of ARP1, MM1.S, and 5TGM1 cells cultured in RPMI 1640 media with or without glycine supplementation (10 mg/L) ($n = 3$ independent experiments; Results represent means ± SD). **d** Schematic of in vivo experiments. **e** Targeted assays of glycine in serum derived from 5TGM1 MM mice fed with control diet ($n = 6$ at week 0, $n = 4$ at week 6) or glycine-free diet ($n = 6$ at week 0 and 6). **f** Tumor-associated luminescence intensity in live 5TGM1 MM mice fed with control diet or glycine-free diet. **g** Quantification of luminescence intensity in 5TGM1 MM mice fed with control diet or glycine-free diet at 2, 4, 6, and 8 weeks. **h** The concentrations of IgG2b in mouse serum as detected with ELISA (Week 0: $n = 6$ in each group, Week 2: $n = 6$ in each group, Week 4: $n = 4$ in control diet group, $n = 6$ in glycine-free group, Week 8: $n = 2$ in control diet group, $n = 6$ in glycine-free group). **i** Micro-CT images of femurs derived from 5TGM1 MM mice fed with control diet or glycine-free diet. **j** Quantification of bone microstructural parameters, namely BV/TV, Tb. N, Tb. Sp, and Tb. Th ($n = 5$ in each group). **k, l** Quantification of bone resorption marker CTX-I and bone formation marker PINP ($n = 4$ in control diet group, $n = 6$ in glycine-free diet group). Results represent means ± SD; Significance was analyzed with an unpaired two-sided $t$-test in (**e, g, h, j–l**). **m** The survival curves of 5TGM1 MM mice fed with control diet or glycine-free diet (Log-rank test). Source data are provided as a Source Data file.

the expression of $\gamma$-$H_2AX$, p-ATR, p-CHK1 and reduced the expression of CDC25A in the tumors, while treatment with GSH restored normal expression levels (Fig. 4n). In addition to glycine cleavage metabolism[19], GLDC is also involved in glycolysis[20–22], purines synthesis[23], GSH synthesis[24,25], autophagy[26], etc. The result of the metabolic flux experiment showed that the proportion of GSH ($m + 1$) in total GSH is much lower than GSH ($m + 2$), indicating that GSH production from the fractional GCS is less than glycine as a direct substrate. We thus consider that glycine cleavage metabolism is just one of the mechanisms by which GLDC promotes GSH synthesis and cell proliferation in MM cells.

**Glycine deprivation enhances the effects of bortezomib on myeloma cells.** We next examined the levels of glycine and glucose in BM and PB in newly diagnosed and relapsed MM patients, finding that the relapsed patients exhibited much lower glycine concentrations but similar glucose concentrations compared to newly diagnosed patients (Supplementary Fig. 5a). While relapsed MM patients had higher intracellular glycine (in MM cells) and lower bone marrow glycine than newly diagnosed patients (Supplementary Fig. 5b). ROC curve analysis suggested that glycine may potentially be useful for distinguishing relapsed from newly diagnosed MM patients (Supplementary Fig. 5c). Interestingly, the addition of glycine, but not glucose, increased the resistance of ARP1 cells to bortezomib (BTZ) (Supplementary Fig. 5d). We also found that the intracellular glycine concentration of ARP1 cells was elevated by BTZ treatment (Fig. 5a). These findings led us to hypothesize that inhibition of glycine utilization may enhance the effects of BTZ on MM cells. To confirm our hypothesis, ARP1 cells cultured with (10 mg/L) or without exogenous glycine were exposed to different doses of BTZ (0, 2.5, 5, 10, or 20 nM) for 48 h, followed by a CCK-8 assay. CCK-8 results showed that glycine deprivation significantly sensitized ARP1 to BTZ (Fig. 5b), while the addition of Trolox (100 μm), a ROS scavenger, reduced the BTZ sensitivity of ARP1 cells cultured without exogenous glycine (Fig. 5c). These findings indicated that exogenous glycine deprivation can sensitize MM cells to BTZ by elevating ROS levels.

We subsequently explored whether glycine deprivation enhances the effect of BTZ on MM in vivo. Thirty 5TGM1 MM mice were randomly and equally divided into five groups, among which two groups were fed with the control diet and the others were fed with the glycine-free diet. After one week, these mice were injected with 5TGM1 cells intravenously. Another week later, groups fed with control diet were injected intraperitoneally three times per week with BTZ (1 mg/kg) or physiological saline, and groups fed with glycine-free diet were injected three times per week with BTZ (1 mg/kg), BTZ plus GSH

(once every other day, 2 mg/kg), or physiological saline. The tumor burden in MM mice was monitored via whole-animal live imaging and IgG2b concentration measurement every two weeks. As shown in Fig. 5d, e, the tumor-associated luminescence intensity in the glycine-free group was lower than that in the control group after BTZ treatment at week 6. Meanwhile, the addition of GSH significantly increased tumor-associated luminescence intensity in the BTZ-treated glycine-free group. Moreover, the concentration of serum IgG2b in the BTZ-treated glycine-free group was lower than that in the BTZ-treated control group but was increased significantly by the addition of GSH (Fig. 5f). CTX-I and PINP serum concentrations were higher in the control mice compared with glycine-free mice, moreover, treatment with GSH significantly elevated CTX-I and PINP serum concentrations in glycine-free mice treated with BTZ (Fig. 5g). In addition, the BTZ + GSH-treated glycine-free group showed a shorter lifetime than the glycine-free group treated with BTZ alone (Fig. 5h).

We also used B-NDG mice to confirm our hypothesis. Fifteen B-NDG mice were injected with ARP1 cells subcutaneously and then randomly and equally divided into five groups as described above. After one week, control groups were treated three times per week with BTZ (1 mg/kg) or physiological saline, and glycine-free groups were treated three times per week with BTZ (1 mg/kg), BTZ plus GSH (2 mg/kg), or physiological saline. BTZ significantly decreased tumor volume in both the control group and glycine-free groups. However, tumors in the glycine-free groups were smaller than those in the BTZ-treated control group, while GSH addition increased the tumor size in the BTZ-treated glycine-free group (Supplementary Fig. 5e). These findings indicate that glycine deprivation enhances the effect of BTZ on MM by decreasing GSH levels in MM cells.

**Blocking glycine uptake inhibits cell proliferation and enhances the effect of BTZ on myeloma cells.** Glycine is an essential neurotransmitter for humans[27] and may therefore not be a feasible therapeutic target. The results from our RNA-sequencing analysis showed that the glycine transporter *SLC6A9* was down-regulated upon glycine deprivation in ARP1 cells, and this was further confirmed with western blotting (Supplementary Fig. 6a). We found that BTZ treatment significantly increased *SLC6A9* expression but did not affect *SLC6A5* expression in ARP1 cells (Fig. 6a). Thus, *SLC6A9* may mediate glycine uptake in MM cells. To test this, we knocked down *SLC6A9* in ARP1 cells (Supplementary Fig. 6b), which caused glycine concentrations and GSH levels to decrease (Fig. 6b, c). *SLC6A9* knockdown also significantly sensitized ARP1 cells to BTZ treatment and increased BTZ-induced ROS levels, while the addition of Trolox reversed

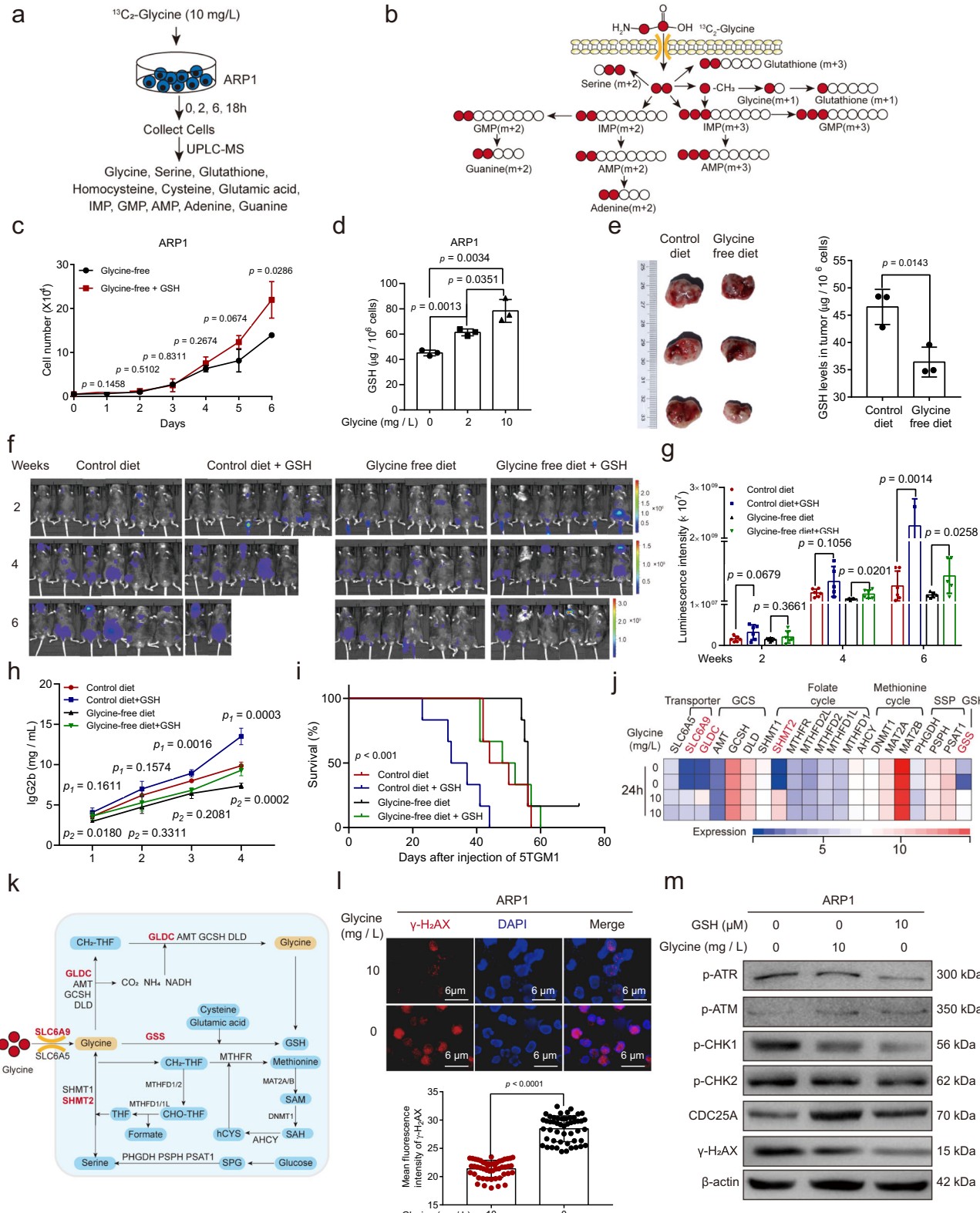

the effect of *SLC6A9* knockdown on both BTZ-induced cell death and ROS levels (Fig. 6d, e, Supplementary Fig. 6c–e).

We subsequently explored the therapeutic potential of the glycine analogues *N,N*-dimethylglycine and betaine, both of which have chemical structures similar to glycine (Fig. 6f). We found that betaine was more effective in reducing the number of viable ARP1 and 5TGM1 cells than *N,N*-dimethylglycine (Fig. 6g) and that betaine treatment reduced both glycine uptake and GSH

levels in MM cells (Fig. 6h, i). To test the therapeutic potential of betaine in vivo, we used 5TGM1 MM mice and divided them into three groups. One week after injection of 5TGM1 cells, the three groups of mice were treated with betaine, BTZ, or physiological saline, respectively. Tumor-associated luminescence intensity increased in all three groups over time (Fig. 6j). However, both betaine and BTZ decreased tumor-associated luminescence intensity and IgG2b concentration in 5TGM1 MM mice at week

**Fig. 3 Glycine-derived GSH contributes to the proliferation of MM cells. a** Schematic of the glycine metabolic flux experiments. **b** Schematic of glycine metabolism in MM cells. **c** Growth curves of ARP1 cells cultured with glycine-free media treatment with or without GSH. **d** GSH levels in ARP1 cells cultured with different doses of glycine for 24 h. $n = 3$ independent experiments; Results represent means ± SD; Unpaired two-sided $t$-test was used in (**c**, **d**). **e** GSH levels in tumor knots derived from B-NDG mice fed with control diet or glycine-free diet ($n = 3$ mice; Unpaired two-sided $t$-test). **f** Live imaging of the tumor-associated luminescence intensity of 5TGM1 MM mice. **g** Quantification of tumor-associated luminescence intensity in the 5TGM1 MM mice cohorts shown in panel (**f**). **h** The concentrations of IgG2b in mouse serum as detected with ELISA (Week 0 and 2: $n = 6$ in each group, Week 4: $n = 5$ in control + GSH group, $n = 6$ in other groups, Week 6: $n = 6$ in control group and glycine-free group, $n = 2$ in control + GSH group, $n = 5$ in glycine-free + GSH group), $p_1$ represent the significance between control group and control + GSH group, $p_2$ represent the significance between glycine-free group and glycine-free + GSH group). Results represent means ± SD; Unpaired two-sided $t$-test was used in (**g**, **h**). **i** The survival curves of 5TGM1 MM mice fed with the control or glycine-free diets with or without GSH ($n = 6$ for each group; Log-rank test). **j** The expression profile of glycine metabolism-related genes in ARP1 cells cultured with or without glycine. **k** Schematic of glycine metabolism. **l** Representative images of the immunofluorescence analysis of γ-H$_2$AX (red) protein expression in ARP1 cells cultured with or without glycine for 48 h ($n = 50$ images in each group; Results represent means ± SD; Unpaired two-sided $t$-test). **m** Western blotting of γ-H$_2$AX, p-ATR, p-ATM, p-CHK1, p-CHK2, CDC25A, and β-actin in ARP1 cells cultured with or without glycine and/or GSH for 48 h. Source data are provided as a Source Data file.

6 (Fig. 6k, l). Betaine and BTZ both also significantly extended mouse survival compared to the control (Fig. 6m).

To explore whether betaine enhances the effect of BTZ on MM cells, we first determined the IC50 values of betaine and BTZ in both ARP1 and 5TGM1 cells (Supplementary Fig. 6f, g) and then assessed the combination index (CI) of betaine and BTZ in ARP1 cells by using the Chou-Talalay method. The CI value of betaine (4 mg/mL) plus BTZ (7 nM) was less than 1 in ARP1 cells, indicating that betaine was synergistic with BTZ in its effect on MM cells (Fig. 6n). Moreover, ARP1 cells treated with the combination of BTZ plus betaine showed higher ROS levels than those treated with the single drugs alone (Fig. 6o). Furthermore, GSH addition markedly reduced the combined effect of betaine and BTZ on cell death induction (Fig. 6p, Supplementary Fig. 6h). The synergy between betaine and BTZ was further evidenced in treated ARP1 cells by elevated γ-H2AX levels and the cleavage of caspase-3 and PARP, which could also be reversed by GSH treatment (Fig. 6q). These findings indicated that betaine decreases GSH levels by blocking glycine uptake and thus increases ROS levels in MM cells, thereby enhancing the effect of BTZ on MM.

**Glycine accumulation in the BM microenvironment is caused by MMP13-induced degradation of bone collagen.** Collagen is composed of amino acids including glycine, L-proline, L-glutamic acid, and L-alanine, among which glycine accounts for about 30% of the total amino acids (Fig. 7a)[28]. Our untargeted metabolomics studies showed that glycine, L-proline, L-glutamic acid, and L-alanine were all elevated in MM patients compared to HDs (Fig. 7b). In addition, MM patients with osteolysis had higher serum glycine levels than MM patients without osteolysis (Fig. 7c), especially for the MM patients in the stage DS-III (Supplementary Fig. 7a, Supplementary Data 5), while lung cancer patients with bone metastases also showed higher serum glycine concentrations than those without bone metastases (Supplementary Fig. 7b). Moreover, the serum concentrations of CTX-I and PINP were positively correlated with serum glycine concentrations in MM patients (Fig. 7d). Together, these results suggested bone collagen as a potential source of glycine in MM patients.

Collagen is degraded by matrix metalloproteinases (MMPs) secreted by tumor cells in various tumor types[29–32]. Consistent with earlier findings[33], we found that ARP1 cells predominantly express *MMP13* (Supplementary Fig. 7c), the concentration of which was positively correlated with both osteolysis (Supplementary Fig. 7d) and glycine concentrations (Fig. 7e) in MM patients. Thus, glycine accumulation was likely caused by MMP13 induction of bone collagen degradation.

To examine the source of glycine accumulation in vivo, we treated 5TGM1 MM mice with CL-82198 (2 mg/kg), a specific

inhibitor of MMP13[34], and then assessed tumor-associated luminescence, bone damage, and glycine concentrations (Fig. 7f). As shown in Fig. 7g–i, CL-82198-treated mice exhibited significantly lower tumor-associated luminescence intensity and higher trabecular bone mass in femurs compared to control mice. Furthermore, quantification of bone microstructural parameters showed that CL-82198 treatment increased femoral BV/TV and Tb. N without changing Tb. Sp and Tb. Th (Fig. 7j). Serum concentrations of CTX-I and PINP in CL-82198-treated mice were significantly lower than those in control mice (Fig. 7k). In addition, glycine concentrations in mice treated with CL-82198 were significantly lower at week 6 than in those treated with saline (Fig. 7l). Thus, MMP13 induction of bone collagen degradation serves as a source for glycine accumulation in the BM.

## Discussion

Our results show that glycine deprivation significantly inhibits MM cell proliferation. We also showed that exogenous glycine is absorbed by MM cells and subsequently metabolized into GSH, purines, and serine, in contrast to a previous publication wherein exogenous glycine was found to only contribute to purine synthesis in melanoma[14]. Mechanistic investigation revealed that glycine deprivation inhibits cell proliferation through the increase of DNA damage and thus cell cycle arrest in MM cells. Moreover, knockdown of GLDC, which has been shown to drive tumor-igenesis in non-small cell lung cancer[21], resulted in decreased GSH, increased DNA damage, and inhibition of proliferation in MM cells. The addition of GSH was effective in reversing these phenotypes caused by glycine deprivation or GLDC knockdown in MM. The metabolic flux analysis of glycine showed that GSH production from GCS is less than glycine as a direct substrate in MM cells. Thus, there may other mechanisms by which GLDC promotes GSH synthersis and cell proliferation. It is well-known that plasma cells account for less than 1% of the total cells in the BM microenvironment in HDs; however, the percentage of malignant plasma cells rises to more than 10% in MM patients[35]. Previous publications from our group and others showed that intracellular GSH decreases the effect of BTZ on MM cells[36,37]. We therefore hypothesize that exogenous glycine is required for MM cells to maintain intracellular GSH balance and antagonize anti-cancer drugs. In this study, we found that glycine deprivation enhanced the effect of BTZ on MM both in vitro and in vivo. Intriguingly, inhibition of glycine utilization via betaine inhibited the proliferation of MM cells and enhanced the anti-cancer effects of BTZ. It has been shown that a high betaine intake is inversely associated with the risk of various cancer types, including color-ectal cancer[38,39], breast cancer[40], liver cancer[41], and prostate cancer[42]. For instance, Kar et al. found that betaine suppresses cell proliferation by increasing oxidative stress-mediated apoptosis in

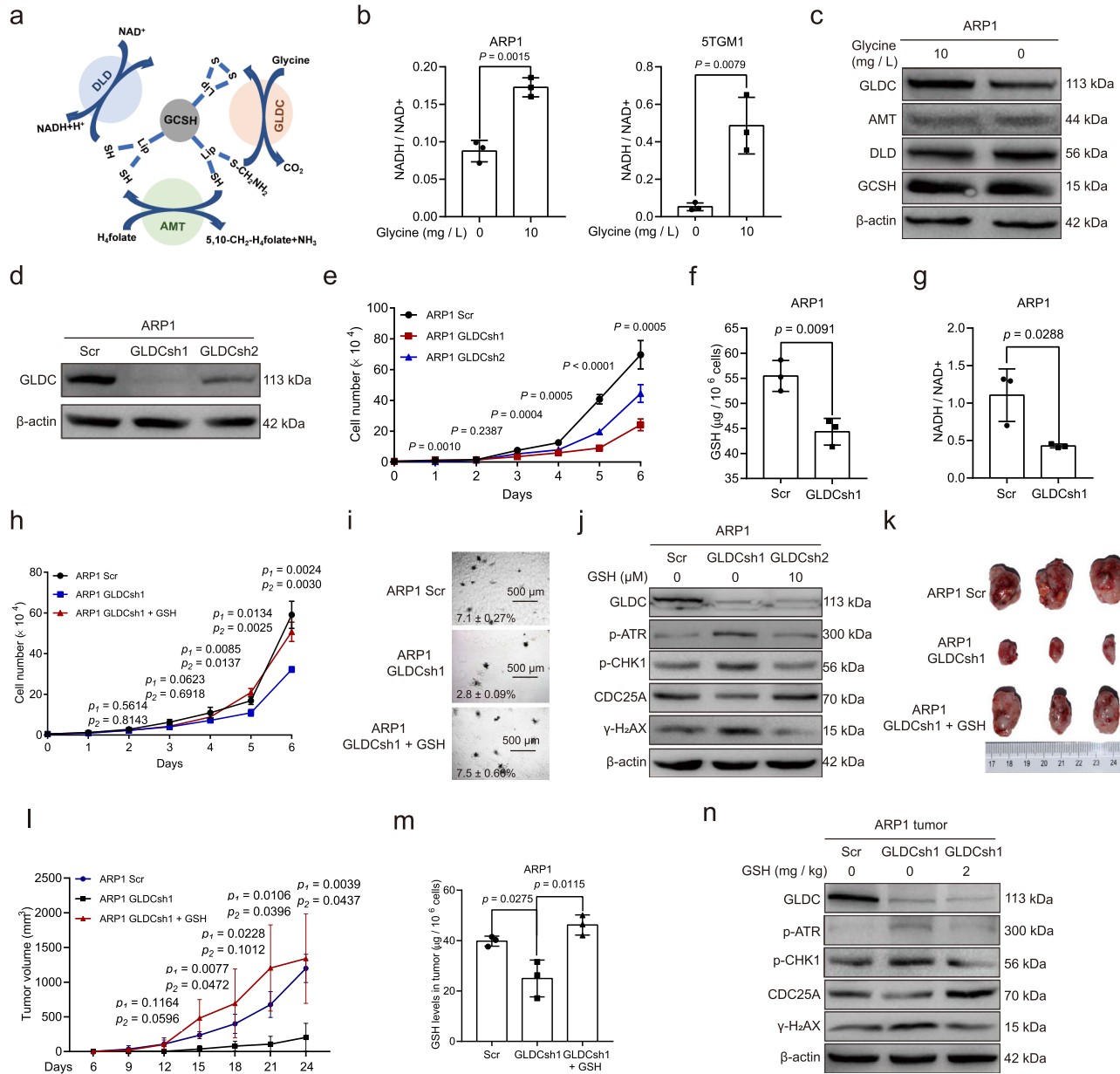

**Fig. 4 Knockdown of GLDC impairs the proliferation of MM cells. a** Schematic of the glycine cleavage system. **b** The ratio of NADH/NAD + in ARP1 and 5TGM1 cells cultured with or without exogenous glycine. **c** Western blotting of GLDC, GCSH, DLD, AMT, and β-actin in ARP1 cells cultured with or without glycine for 24 h. **d** Western blotting of GLDC and β-actin in ARP1 cells with Scramble (Scr), GLDC shRNA1 (GLDCsh1) and GLDCsh2. **e** Growth curves of ARP1 cells with Scr, GLDCsh1, or GLDCsh2. **f** GSH levels in ARP1 cells with Scr and GLDCsh1. **g** The ratio of NADH / NAD+ in ARP1 cells with Scr or GLDCsh1. **h** Growth curves of ARP1 cells with Scr or GLDCsh1 with or without GSH treatment. **i** Clonogenic analysis of ARP1 cells with Scr or GLDCsh1 with or without GSH treatment. $n = 3$ independent experiments; Results represent means ± SD; Significance was analyzed with unpaired two-sided $t$-test in (**b, f–h**) ($p_1$ represent the significance between Scr and GLDCsh1, $p_2$ represent the significance between GLDCsh1 and GLDCsh1 + GSH), with ANOVA one-way test in (**e**). **j** Western blotting of GLDC, p-ATR, p-CHK1, CDC25A, γ-H2AX, and β-actin in ARP1 cells with Scr or GLDCsh1 with or without GSH. **k** Tumor knots developed in B-NDG mice injected with ARP1 cells with Scr ($n = 3$ mice), GLDCsh1 without GSH treatment ($n = 3$ mice), or GLDCsh1 cells with GSH treatment (2 mg/kg) ($n = 3$ mice). **l** Analysis of tumor volumes in Scr, ARP1 GLDCsh1, and ARP1 GLDCsh1 + GSH mice. **m** GSH levels in tumor knots derived from Scr, ARP1 GLDCsh1, and ARP1 GLDCsh1 + GSH mice. $n = 3$ mice; Results represent means ± SD; Significance was analyzed with an unpaired two-sided $t$-test in (**l**) ($p_1$ represent the significance between Scr and GLDCsh1, $p_2$ represent the significance between GLDCsh1 and GLDCsh1 + GSH), **m. n** Western blotting of GLDC, p-ATR, p-CHK1, CDC25A, γ-H2AX, and β-actin in tumor knots derived from Scr, ARP1 GLDCsh1, and ARP1 GLDCsh1 + GSH mice. Source data are provided as a Source Data file.

prostate cancer cells[43]. These reports, along with our current study, led us to conclude that betaine in combination with BTZ may serve as a promising strategy for MM therapy.

Bone collagen is one of the most important components of the bone matrix and can be degraded by MMPs in multiple cancer types, contributing to bone metastasis and tumor progression[32]. MM cells induced high levels of MMP13 expression, promoting osteolysis and reducing overall survival in MM[33,44]. Previous studies showed that osteolysis was associated positively with the elevation of serum glycine[45]. In the present study, we found that

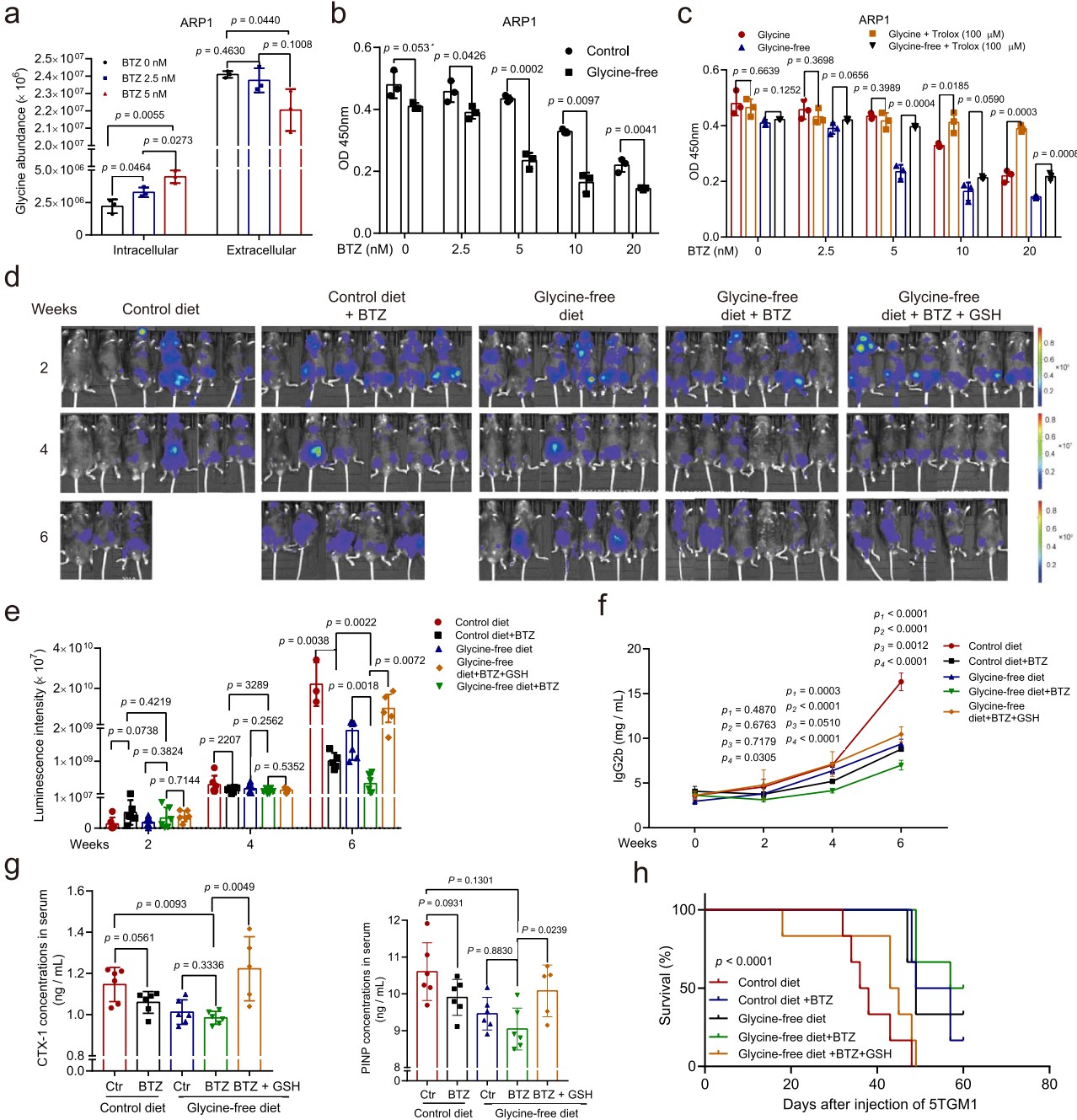

**Fig. 5 Glycine deprivation sensitizes MM cells to BTZ. a** Intracellular and extracellular glycine concentrations in ARP1 cells treated with different doses of BTZ (0, 2.5, or 5 nM) for 24 hr. **b** ARP1 cells cultured with (10 mg/L) or without exogenous glycine were treated with different doses of BTZ (0, 2.5, 5, 10, or 20 nM) for 48 h, followed by a CCK-8 assay. **c** ARP1 cells cultured with (10 mg/L) or without exogenous glycine were treated with different doses of BTZ (0, 2.5, 5, 10, or 20 nM) and with or without Trolox (100 μM) for 48 hr, followed by a CCK-8 assay. $n = 3$ independent experiments; Results represent means ± SD; Significance was analyzed with an unpaired two-sided nonparametric $t$-test in (**a**–**c**). **d** Live imaging of tumor-associated luminescence intensity in 5TGM1 MM mice in the groups fed with the control diet ($n = 6$ mice), fed with control diet and treated with BTZ (1 mg/kg, 3 times/week, $n = 6$ mice), fed with glycine-free diet ($n = 6$ mice), fed with glycine-free diet and treated with BTZ (1 mg/kg, 3 times/week, $n = 6$ mice), or fed with glycine-free diet and treated with BTZ (1 mg/kg, 3 times/week) plus GSH (2 mg/kg, 3 times/week) ($n = 6$ mice). **e, f** The luminescence intensities and serum IgG2b concentrations of the five groups of 5TGM1 MM mice. **g** Quantification of bone resorption marker CTX-I and formation marker PINP. Results represent means ± SD; Significance was analyzed with an unpaired two-sided nonparametric $t$-test in **e, f** ($p_1$ represent the significance between contro and control + BTZ, $p_2$ represent the significance between glycine-free and glycine-free + BTZ, $p_3$ represent the significance between control + BTZ and glycine-free + BTZ, $p_4$ represent the significance between glycine-free + BTZ and glycine-free + BTZ + GSH), **g**. **h** The survival curves of the five groups of 5TGM1 MM mice (Log-rank test). Source data are provided as a Source Data file.

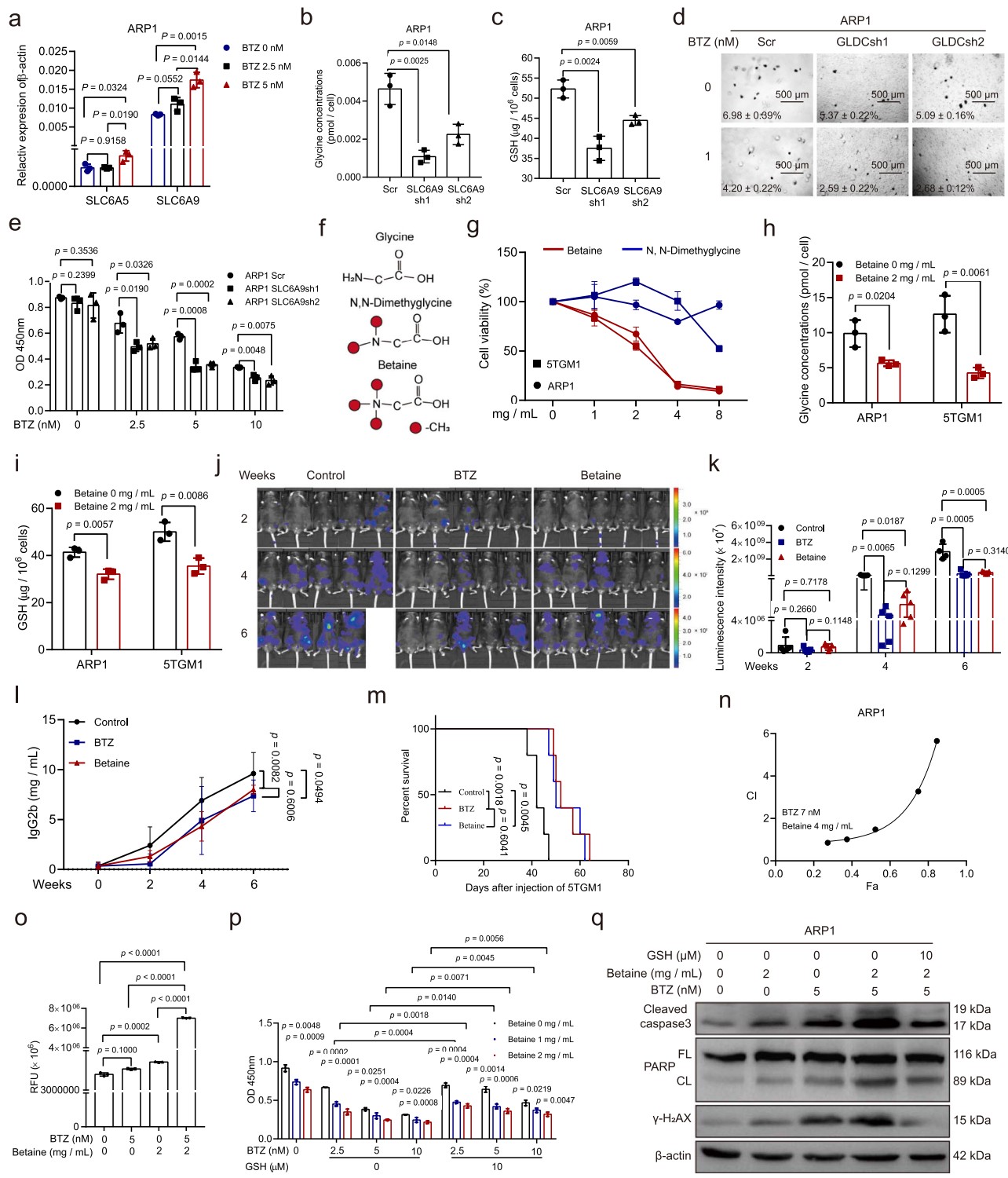

increased glycine concentrations in the serum are linked to osteolysis in MM and lung cancer. We also showed that the serum MMP13 concentration correlates with the serum glycine concentration and osteolysis in MM patients. Importantly, the treatment of 5TGM1 MM mice with an MMP13 inhibitor reduced the serum glycine concentration and inhibited tumor progression. Therefore, our data support the theory that the elevation of glycine in the BM microenvironment of MM patients is caused by the degradation of bone collagen mediated by MM cell-secreted MMP13. Glycine may also contribute to bone

metastasis and osteolysis-mediated progression of multiple cancer types. This potential mechanism should be further examined in the future.

In conclusion, we show that elevation of glycine in the BM contributes to MM progression by modulating the balance of GSH in MM cells. Inhibition of glycine utilization by betaine treatment exhibits strong therapeutic effects on MM, either alone or in combination with BTZ (Fig. 7m). Taken together, our findings highlight the mechanistic role of glycine in MM progression and reveal a therapeutic strategy for MM by targeting glycine metabolism.

**Fig. 6 Inhibiting glycine utilization via *SLC6A9* knockdown or betaine treatment reduces proliferation and enhances the effect of BTZ on MM cells.**
**a** *SLC6A5* and *SLC6A9* expression in ARP1 treated with BTZ. **b** Glycine concentrations in ARP1 Scr, ARP1 SLC6A9sh1, and ARP1 SLC6A9sh2 cells. **c** GSH levels in ARP1 Scr, ARP1 SLC6A9sh1, and ARP1 SLC6A9sh2 cells. **d** Clonogenic analysis of ARP1 Scr, ARP1 SLC6A9sh1, and ARP1 SLC6A9sh2 cells with or without BTZ treatment. **e** CCK-8 assay of ARP1 Scr, ARP1 SLC6A9sh1, and ARP1 SLC6A9sh2 cells treated with different doses of BTZ. **f** The chemical structures of glycine, *N,N*-Dimethylglycine, and betaine. **g** Cell viability of ARP1 and 5TGM1 treated with *N,N*-Dimethylglycine or betaine. **h** Glycine levels in ARP1 and 5TGM1 cells with or without betaine treatment. **i** GSH levels in ARP1 and 5TGM1 with or without betaine treatment. **j** Live imaging luminescence intensity in 5TGM1 MM mice treated with physiological saline, BTZ (1 mg/kg), or betaine (500 mg/kg) ($n = 5$ for each group). **k** Quantification of tumor-associated luminescence intensity in 5TGM1 MM mice at 2 ($n = 5$ in each group), 4 ($n = 5$ in each group), and 6 ($n = 4$ in control group, $n = 5$ in other groups) weeks. **l** IgG2b concentrations in 5TGM1 MM mice treated with physiological saline, BTZ, or betaine. **m** The survival curves of 5TGM1 MM mice (Log-rank test). **n** The combination index of betaine and BTZ treatment in ARP1 was analyzed by the Chou–Talalay method. **o** The ROS levels in ARP1 cells with or without BTZ and betaine treatment. **p** CCK-8 assays of ARP1 treated with different doses of BTZ and with or without betaine and GSH. **q** Western blotting of cleaved caspase3, PARP, γ-H$_2$AX, and β-actin in ARP1 and 5TGM1 with or without BTZ, betaine and GSH. $n = 3$ independent experiments; Results represent means ± SD in (**a–e, g–i, o, p**). Unpaired two-sided *t*-test was used in (**a–c, e, h, i, k, o, p**); paired two-sided *t*-test was used in was applied in (**l**). Source data are provided as a Source Data file.

## Methods

**Human subjects.** We confirm that this study was approved by the Cancer Research Institute Review Board of Central south university (CSU). The BM and PB specimens were derived from HDs ($n_{BM} = 52$; $n_{PB} = 57$), newly diagnosed MM patients ($n_{BM} = 160$; $n_{PB} = 141$), relapsed MM patients ($n_{BM} = 34$; $n_{PB} = 34$), and patients with lung cancer ($n_{PB} = 38$) with written consent from Xiangya Hospital, the Second, the Third Xiangya Hospital of CSU and the Blood Diseases Hospital of Chinese Academy of Medical Science & Peking Union Medical College during the period 2014–2019. Blood samples and bone marrow samples were collected from all subjects, and plasma, sera and bone marrow supernatants were extracted from these specimens and stored at −80 °C until analysis. Informed consent was obtained in accordance with the Declaration of Helsinki.

**Cell lines, antibodies, and reagents.** The human B cell line GM12878, the human MM cell lines ARP1, MM.1S, KMS28-PE, OCI-My5, and RPMI-8226, and the mouse MM cell line 5TGM1 were cultured in Roswell Park Memorial Institute (RPMI) 1640 medium (Gibco, USA) supplemented with 10% fetal bovine serum (FBS; Gibco). Reagents used in the study included glycine (#G8790, Sigma-Aldrich, USA), $^{13}C_2$-glycine (#283827, Sigma-Aldrich), Bortezomib (Janssen, China), Betaine (#B2629, Sigma-Aldrich), Reduced glutathione (#MB3281, Meilunbio, China), L-Buthionine-(S,R)-sulfoximine (#HY-106376A, MedChemExpress, USA), CL-82198 (#HY-100359, MedChemExpress), and L-amino acids (#LAA21, Supelco, USA). The antibodies used in this study were purchased as follows: anti-GLDC (#24827-1-AP, 1: 1000), anti-DLD (#16431-1-AP, 1: 1000), anti-GCSH (#16726-1-AP, 1: 1000), anti-phosphorylated CDC25A (#55031-1-AP, 1: 1000), and anti-β-actin (#66009-1-Ig, 1:5000) were all purchased from Proteintech Group. Anti-AMT (#A9926, 1: 1000) and anti-SLC6A9 (#A16203,1: 1000) were purchased from ABclonal. Anti-phosphorylated ATR (#2853S, 1: 1000), anti-phosphorylated ATM (#5883S, 1: 1000), anti-phosphorylated CHK1 (#2348S, 1: 1000), anti-phosphorylated CHK2 (#2197S, 1:1000), and anti-γ-H$_2$AX (#9718S, 1: 1000 for western blot, 1: 100 for immunofluorescence) were obtained from Cell Signaling Technology. HRP-conjugated goat anti-Rabbit IgG (#L3042, 1: 5000) and Goat anti-Mouse IgG (#101, 1: 5000) secondary antibodies were purchased from Signalway Antibody.

**Untargeted metabolomics assays.** The untargeted metabolomics profiling was performed on XploreMET platform (Metabo-Profile, China). The sample preparation procedures are referred to in their previously published methods with minor modifications[46]. Firstly, the plasma samples and BM liquid samples derived from HDs and MM patients were put on ice and then centrifuged for 5 min at 4 °C and 3000 g (Microfuge 20R, Beckman Coulter, Inc, USA) to separate debris or a lipid layer. Each aliquot of 50 μL sample was mixed with 10 μL of internal standard, to which 175 μL of pre-chilled methanol/chloroform (v/v = 3/1) was added. After centrifugation at 14,000 g and 4 °C for 20 min (Microfuge 20R, Beckman Coulter, Inc), each 200 μL of the supernatant was carefully transferred to an autosampler vial (Agilent Technologies, USA). The remaining supernatant from each sample was pooled to make quality control samples. All the samples in autosampler vials were evaporated briefly to remove chloroform using a CentriVap vacuum concentrator (Labconco, USA), and further lyophilized with a FreeZone freeze dryer equipped with a stopping tray dryer (Labconco).

The sample derivatization and injection were performed by a robotic multipurpose sample MPS2 with dual heads (Gerstel, Germany). Briefly, the dried sample was derivatized with 50 μL of methoxyamine (20 mg/mL in pyridine) at 30 °C for 2 h, followed by the addition of 50 μL of MSTFA (1% TMCS) containing FAMEs as retention indices at 37.5 °C for another 1 h using the sample preparation head. In parallel, the derivatized samples were injected with a sample injection head after derivatization. Every sample is introduced for analysis within 6 h after it has been prepared. A time-of-flight mass spectrometry (GC-TOF/MS) system (Pegasus HT, USA) with an Agilent 7890B gas chromatography and a Gerstel multipurpose

sample MPS2 with dual heads (Gerstel) was used to investigate the metabolite profiling in HDs and MM patients. A Rxi-5 ms capillary column (30 m × 250 μm i.d., 0.25-μm film thickness; Restek corporation, USA) was used for the separation of metabolites. Helium was used as the carrier gas at a constant flow rate of 1.0 mL/min. The temperature of injection and transfer interface were both set to 270 °C. The source temperature was 220 °C. The measurements were made using electron impact ionization (70 eV) in the full scan mode (*m/z* 50–500). A comprehensive set of rigorous quality control/assurance procedures is employed to ensure a consistently high quality of analytical results, throughout controlling every single step from sample receipt at laboratory to final deliverables. Sample derivatization in parallel with ongoing GC-TOF/MS analysis are performed on a robotic multipurpose sample with dual heads (Gerstel).

Metabolite annotation was performed by comparing the retention indices and mass spectral data with those previously generated from reference standards of known structures present in JiaLib metabolite database using ChromaTOF software. The resulting datasets were then analyzed by using the SIMCA-P 14.1 software package. OPLS-DA was carried out to visualize the metabolic alterations between HDs and MM patients. The VIP value generated in OPLS-DA models ranked the overall contribution of each metabolite to distinguishing various groups, and those metabolites with VIP > 1.0 were considered responsive for group discrimination. In parallel, annotated metabolites were analyzed at the univariate level by using MetaboAnalyst 5.0 online software (https://www.metaboanalyst.ca/) with unpaired two-sided nonparametric *t*-test[47]. Birfly, the data of metabolites abundances was uploaded onto the MetaboAnalyst 5.0 software through Statistical Analysis (one factor) module and then followed by data integrity check, data filtering, sample normalization, data Log transformation, and data scaling. Ultimately, the FC and *p*-value were analyzed by unpaired two-sided nonparametric *t*-test.

**Targeted metabolomics assays.** Measurements of glycine in the cell pellet, conditioned media, and mouse serum were performed by using high-performance liquid chromatography (HPLC). Serum samples were prepared as follows: a 10 μL aliquot of serum was mixed with 20 μL of 1% salicylic acid solution. The mixture was vortexed for 1 min, allowed to stand for 20 min on ice, and then centrifuged at 20,000 g at 4 °C for 10 min. The supernatant was transferred to a clean tube for HPLC analysis. Cell samples were prepared as follows: 80 μL hydrochloric acid (10 mM) was added into the cell pellet, mixed, and frozen and thawed twice, followed by the addition of 20 μL 5% salicylic acid solution. The mixture was incubated on ice for 10 min and then centrifuged at 8000 g at 4 °C for 5 min. The supernatant was transferred to a clean tube for HPLC analysis. The derivatizing agent was prepared by mixing 0.5 mL methanol, 10 mg o-Phthalaldehyde (OPA; Sigma-Aldrich), 2 mL boric acid buffer (0.4 M), and 30 μL mercaptoethanol (Sigma-Aldrich), and then stored at 4 °C. Mobile phase A was prepared with 50 mM sodium acetate, and its pH value was adjusted to 6.8 by the addition of glacial acetic acid. Mobile phase B was prepared by mixing 275 mL chromatographic methanol (Sigma-Aldrich) and 12.5 mL tetrahydrofuran (Sigma-Aldrich). A 10 μL sample was mixed with 100 μL derivatizing agent and allowed to stand for 3 min at room temperature. The mixture was then injected into an HPLC system (Agilent, USA) with an Ultimate XB-C18 chromatographic column (Welch, China). The column was held at 30 °C. The elution procedure for the column was 20% B for 0.02 min, 25% B for 6 min, 37% B for 16 min, 59% B for 25 min, 78% B for 29 min, 78% B for 32 min, 25% B for 35 min, 25% B for 40 min. The flow rate was 1 mL/min. The data of HPLC was collected by using LabSolutions CS software (Shimadzu, Japan). Measurements of glycine and glucose in serum and bone marrow supernatants were performed by using the Q300 Metabolite Assay Kit (Human Metabolomics Institute, Inc, China) as previously published[48].

**Isotope tracing.** Isotope tracing was performed based on the previous publications[14,49,50]. ARP1 cells were washed twice with phosphate-buffered saline

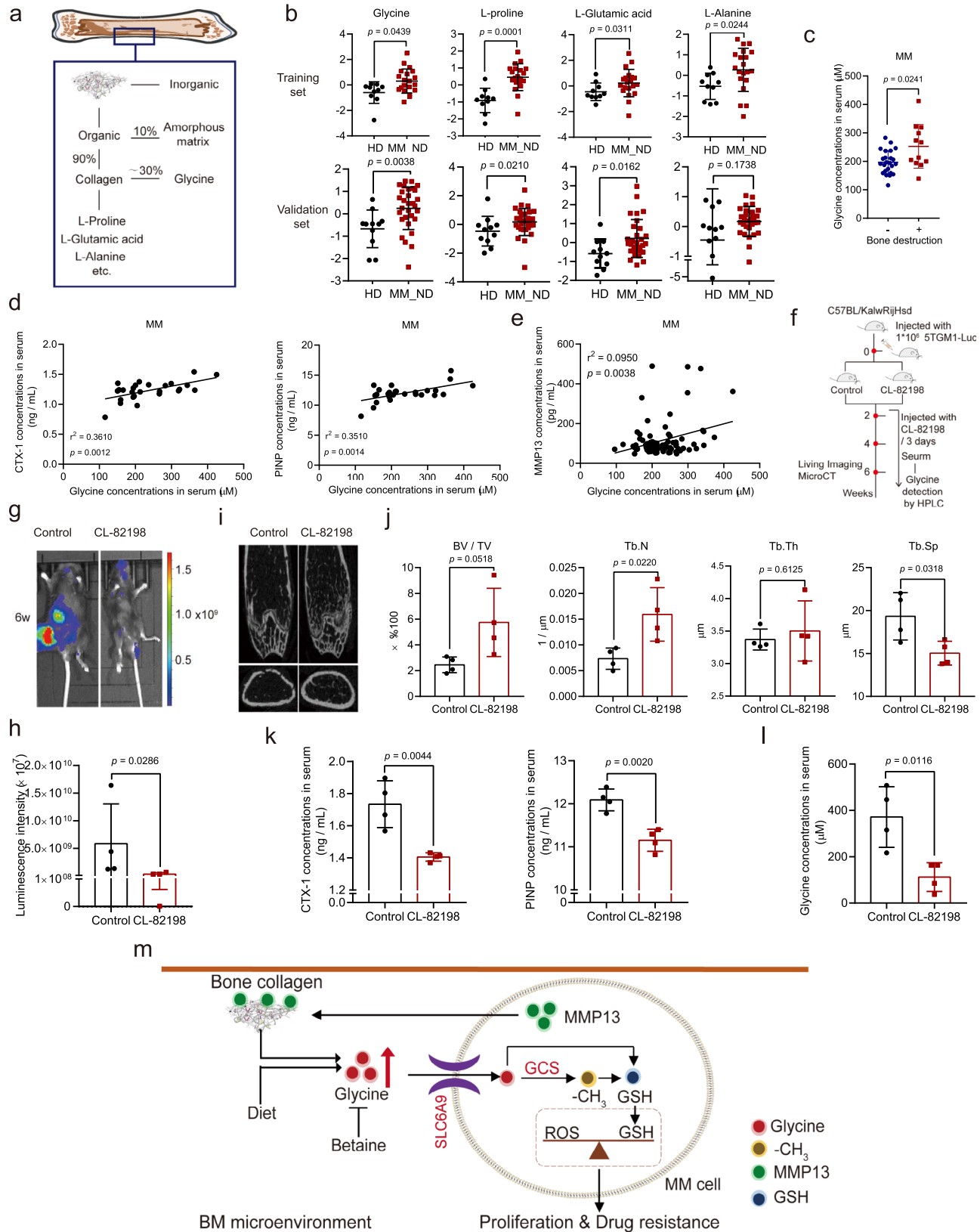

(PBS). 1 × 10[7] cells were then seeded in a T75 flask in glycine-free Roswell Park Memorial Institute (RPMI) 1640 media (BOSTER, China) supplemented with 5% dialyzed fetal bovine serum (FBS; Biological Industries, Israel) and 10 mg/L $^{13}C_2$-glycine (Sigma-Aldrich). Cells were collected at 0, 2, 6, and 18 h. Sample collected at 0 h were used as a reference for the analysis. Cell pellets were washed three times with cold PBS, and the dry pellets were stored at −80 °C for subsequent analysis.

After correcting for the natural isotope (see below), samples were analyzed using ultra-performance liquid chromatography triple-quadrupole mass spectrometry (UPLC-TQ-MS) (Waters Corp, UK) following our previously published procedure. Metabolites were separated through a 2.1 × 100 mm 1.7 μm Acquity amide and an Acquity HSS C18 column (Waters Corp) equipped with an ACQUITY UPLC VanGuard Pre-Column (Waters Corp). A 5 μL aliquot of the sample was injected into the column, which was maintained at 40 °C. The flow rate

**Fig. 7 Bone collagen degradation, mediated by MM cell-secreted MMP13, increases glycine levels in MM patients. a** Schematic of the components of the bone matrix. **b** Untargeted metabolomics assays of glycine, L-proline, L-glutamic acid, and L-alanine in BM liquid derived from MM patients ($n = 20$ in training set group, $n = 30$ in validation set group;) and HDs ($n = 10$ in training set group, $n = 11$ in validation set group). **c** Serum glycine concentrations in MM patients with ($n = 12$) or without ($n = 24$) bone destruction. **d** The correlations between serum glycine concentrations and serum CTX-I or PINP in MM patients ($n = 26$). **e** The correlation between serum MMP13 and serum glycine concentrations in MM patients ($n = 87$). Two-sided Pearson correlation analysis was applied in (**d**, **e**). **f** Schematic of in vivo experimental workflow. **g** Tumor-associated luminescence intensity in 5TGM1 MM mice treated with physiological saline (Control) or CL-82198 (2 mg/kg) ($n = 4$ for each group). **h** Quantification of the tumor-associated luminescence intensity shown in panel **g** ($n = 4$ for each group). **i** Micro-CT images of femurs derived from 5TGM1 MM mice fed with control diet or glycine-free diet. **j** Quantification of bone microstructural parameters, including BV/TV, Tb. N, Tb. Sp, and Tb. Th ($n = 4$ for each group). **k** Quantification of serum CTX-I and PINP in mice with or without CL-82198 treatment ($n = 4$ for each group). **l** Glycine concentrations in serum derived from 5TGM1 MM mice treated with physiological saline or CL-82198 (2 mg/kg) ($n = 4$ for each group). **m** Schematic of our working hypothesis. Results represent means ± SD; Significance was analyzed with an unpaired two-sided $t$-test in (**b**, **c**, **j**, **h**, **k**, **l**). Source data are provided as a Source Data file.

remained constant at 0.4 mL/min. The raw UPLC-MS data were analyzed using TargetLynx Application Manager software version 4.1 (Waters Corp). Quantification of each metabolite was performed by using linear regression analysis of the peak area of metabolite versus concentration.

The measurements of mass distribution vector (MDV) of [13]C-labeled metabolites, which describes the fractional abundance of each isotopologue normalized to the sum of all possible isotopologues, were performed in all detected metabolites. Before calculation of the MDVs of metabolites, the correction of naturally occurring isotopes was performed based on the previous publication[50]. At first, the proportion ($P_0$) of natural [13]C-labeled metabolite relative to non [13]C-labeled metabolite in ARP1 cells collected at 0 h was calculated. $P_0$ = abundance of natural [13]C-labeled metabolite $_{0h}$/abundance of non [13]C labeled metabolite $_{0h}$. The abundance of natural [13]C-labeled metabolite in ARP1 cells cultured with [13]C$_2$-glycine was calculated by the following formula: abundance of natural [13]C-labeled metabolite = abundance of non [13]C-labeled metabolite × $P_0$. Then the abundance of natural [13]C-labeled metabolite was subtracted from total [13]C-labeled metabolite in ARP1 cells cultured with [13]C$_2$-glycine. The MDVs of [13]C-labeled metabolites were obtained by the following formula: MDVs = (abundances of [13]C-labeled metabolites abundances of natural [13]C-labeled metabolites)/(abundances of total metabolites - abundances of natural [13]C-labeled metabolites).

**Western blotting.** Proteins were extracted by using radio immunoprecipitation assay (RIPA) lysis buffer (Beyotime, China) and then quantified using a BCA Protein Quantification Kit (#E112-01, Vazyme, China). Proteins were separated by sodium dodecyl sulfate polyacrylamide gel electrophoresis (SDS-PAGE) and transferred to 0.45 μm polyvinylidene fluoride (Millipore, USA). The membrane was blocked using 5% nonfat dry milk in Tris-buffered saline solution containing 0.1% Tween 20 (TBS-T) for 1 h at room temperature and then probed with specific primary antibodies overnight at 4 °C. Subsequently, The membranes were washed with TBS-T, followed by incubating with HRP-conjugated secondary antibodies for 1 h at room temperature. β-actin was used as the protein loading control. Protein signals were developed with SuperSignalTM West Femto Maximum Sensitivity Substrate (Thermo Fisher Scientific, USA) and imaged using a chemiluminescence imaging system MiniChemi™ 610 with SageCapture TM software version 2.17.12.170316 (Sagecreation, China).

**Quantitative real-PCR.** Total RNA was extracted using Trizol (#15596026, Thermo Fisher Scientific) according to the manufacturer's instructions. Total RNA was retrotranscribed using the SuperScript™ II Reverse Transcriptase kit (#18064071, Thermo Fisher Scientific). Real-time quantitative PCRs (qPCR) were performed by using ABsolute qPCR SYBR Green Mixes (#AB1163A, Thermo Fisher Scientific) according to the manufacturer's instructions. Real-time PCR was run on CFX Real-Time PCR Detection System with CFX Manager TM software, version 3.1 (Bio-Rad). Fold changes were calculated using the ΔΔCt method and *ACTB* mRNA as reference. Primers were obtained from Tsingke Biotechnology Co, Ltd. Primer sequences are listed in Supplementary Table 2. Each sample was repeated three times.

**Micro-CT scanning.** Micro-CT scanning was performed according to a previous publication[51]. Mice femurs were fixed in 4% paraformaldehyde (PFA) for 48 h and stored in 75% ethanol at 4 °C. Femur scans were performed by using High-resolution μCT (Skyscan 1176, Skyscan, Belgium) at a resolution of 8.88 μm per pixel, with a voltage of 50 kV and current of 400 μA. The region of interest (ROI) was selected starting from 0.45 mm below the distal growth plate and extended for 0.45 mm proximally to measure the bone parameters, including trabecular bone volume fraction (Tb. BV/TV), trabecular thickness (Tb. Th), trabecular separation (Tb. Sp) and trabecular number (Tb. N). These bone parameters were analyzed by using CTAn version 1.11 (Bruker, Germany) and μCTVol version 2.2 (Bruker).

**Mouse models of MM.** 6–8-week-old C57BL/KalwRijHsd mice (Strain: C57BL/KaLwRijHsd; Harlan Mice, Netherlands) and 6–8-week-old female B-NDG mice (Strain: NOD.CB17-PrkdcscidIl2rgtm1/Bcgen; Genetic background: Prkdc (−/−), IL2rg (X-/X-); Biocytogen Co, Beijing, China) were used in this study. All mice were maintained under SPF conditions in a controlled environment of 20–22 °C, with a 12/12 h light/dark cycle, 50–70% humidity. All animal experiments were performed in accordance with the guidelines of the Institutional Animal Care and Local Veterinary Office and Ethics Committee of the CSU and the Hunan Normal University, China (animal experimental license number for B-NDG mice: NO.2019sydw0154 and for C57BL/KalwRijHsd mice: D2020001). The 5TGM1 MM mouse model was developed by intravenously injecting luciferase-expressing 5TGM1 cells ($1 \times 10^6$ cells in 200 μL PBS supplemented with 10% FBS) into C57BL/KalwRijHsd mice. MM progression in the mice was monitored by measuring the tumor burden via live animal imaging, serum IgG2b concentration measurement, and micro-CT scans of the femurs. Survival curves were produced for all 5TGM1 MM mice. The xenograft mouse model of MM was prepared by injecting $1 \times 10^6$ ARP1 cells into the right abdomen of immunocompromised B-NDG mice, as previously described in our and others' publications[2,52–55]. The tumor burden was monitored by following the volume of the tumor. The maximal tumor size is 2 cm, which was permited by the Institutional Animal Care and Local Veterinary Office and Ethics Committee of the CSU. We confirm that the maximal tumor size was not exceeded in our study. Control diet and glycine-free diet were prepared by a company (#TP20191211001, Trophic Animal Feed High-Tech Company, Jiangsu, China). The control diet was composed of essential (9.6%) and non-essential amino acids (2.3% glycine and 6% other non-essential amino acids), corn starch (40%), dextrin and sucrose (24.5%), cellulose (5%), soybean oil (7%), minerals (4.2%), vitamins (1%), choline chloride (0.25%), tert-butylhydroquinone (0.0014%), and other components. The glycine-free diet was the same as the control diet, but without glycine, and the other non-essential amino acid levels (8.3%) were increased proportionally to achieve the same total amino acid content. After 1-week injection of MM cells, treatments were performed on mice. Mice were treated with GSH (2 mg/kg) injected intraperitoneal once every other day. BTZ (1 mg/kg) and Betaine (500 mg/kg) were injected intraperitoneal three times every week. Mice treated with physiological saline were set as control.

**Cell proliferation and viability assay.** To assay cell growth, MM cells were washed twice with PBS and plated onto 24-well plates at a density of 5,000 cells per well. Cell numbers were calculated by using a cell counting chamber every day for six days. To assay cell viability, MM cells were plated onto 96-well plates at a density of 5000 cells per well. MM cells were treated with or without drugs for two days, and cell viabilities were then measured by using a Cell Counting Kit-8 (#B34302, Bimake, USA) according to the manufacturer's instructions. Each test was repeated three times.

**Soft-agar clonogenicity assay.** Firstly, we mixed 3.5% low melting point agarose (#16520050, Thermo Fisher Scientific) and media (the volume ratio of agarose and media is 1: 5) to prepare the bottom soft-agar and then added it into 12-wells plates quickly (1 mL soft-agar/well). After the bottom soft-agar has solidified, the upper soft-agar was prepared through mixing 1.66% low melting point agarose and media containing MM cells (the volume ratio of agarose and media is 1: 5). Upper soft-agar was added onto the surface of bottom soft-agar (0.5 mL soft-agar/well, the cell densities in the well as follows: 2500 cells/well in normal media or 5000 cells/well in glycine-free media). Cells were fed with media supplemented with or without drugs twice every week. One colony was defined if more than 40 cells were observed. Plates were imaged under inverted microscope, and colonies were enumerated using ImageJ software. Each sample was repeated three times.

**Vectors, transfections, and transductions.** Short hairpin RNA (shRNA) sequences targeting human *SLC6A9*, *GLDC*, *AMT*, *GCSH*, and *DLD* were obtained from the RNAi consortium collection (MISSION shRNA; Sigma-Aldrich, www.sigmaaldrich.com). shRNA sequences were annealed and ligated into a pLKO-tet-

on lentiviral vector. Lentiviruses were packaged in HEK293T cells using pMD2G and psPAX2 helper vectors and polybrene (3 µg/mL)—mediated transduction (#H9268-5G, Sigma-Aldrich). Transient transfection was performed using Lipofectamine 3000 reagent (#L3000015, Thermo Fisher Scientific) according to the manufacturer's instructions. MM cells transduced with recombinant lentivirus were selected with 1–2 µg/mL puromycin (#A1113803, Thermo Fisher Scientific). shRNA sequences were obtained from Tsingke Biotechnology Co, Ltd. All shRNA sequences are listed in Supplementary Table 3.

**Determination of ROS, GSH, NADH/NAD+, MMP13, CTX-1, and PINP levels**.
ROS level in MM cells was measured by using ROS kit (#MAK143-1KT, Sigma-Aldrich) according to the manufacturer's instructions. MM cells were harvested and plated into 96-well plates (20,000 cells/well). Centrifuged the plate at 200 g for 2 min with brake off prior to the experiment. Add 100 mL/well (96-well plate) of Master Reaction Mix into the cell plate. Incubated the cells in a 5% $CO_2$, 37 °C incubator for 1 h. ROS level in MM cells was measured by detecting the fluorescence intensity at $\lambda_{ex} = 490$ nm/$\lambda_{em} = 525$ nm.

$1 \times 10^7$ MM cells were collected via centrifugation (200 g for 5 min at 4 °C). Cells were washed twice with PBS followed by removal of the supernatant and resuspension in 80 µL of 10 mM HCl. Cells were lysed via two freeze/thaw cycles followed by the addition of 20 µL of 5% 5-Sulfosalicylic acid (5-SSA) and centrifugation at 8000 g for 10 min at 4 °C. The supernatant was moved to a new tube. SSA was diluted to 0.5% with ddH$_2$O. Finally, the intracellular GSH concentration was measured using a glutathione disulphide (GSSG) / GSH quantification kit II (#G257, Dojindo, Japan) according to the manufacturer's instructions.

The intracellular NADH/NAD + level was measured using the NADH/NAD + assay kit with WST-8 (#S0175, Beyotime, China) following the manufacturer's instructions. $1 \times 10^6$ MM cells were collected via centrifugation (200 g for 5 min at 4 °C) and then removed medium. Cells were lysed by the addition of 200 µL pre-cooling NADH/NAD + lysis buffer and centrifugation at 12,000 g for 10 min at 4 °C. 50–100 µL supernatant was moved into a tube followed by 30 min of incubation at 60 °C, and then performed centrifugation at 10,000 g for 5 min at 4 °C. 20 µL supernatant was moved into 96-well plates and incubated 10 min at 37 °C then followed by the addition of 10 µL color-developing solution. The ratio of NADH/NAD+ was measured by detecting the absorbance of mixed liquid at 450 nm.

The MMP13 concentration in the serum of MM patients was measured by using the human MMP-13 enzyme-linked immunosorbent assay (ELISA) kit (#SEKH-0259-96T, Solarbio, China), following the manufacturer's instructions. Added 100 µL serum into the plate and then incubated plate for 90 min at 37 °C. Washed plate with PBST for 4 times, and then added 100 µL biotin-conjugated antibody into plate followed by 60 min of incubation at 37 °C. Washed plate for 4 times, added 100 µL enzyme conjugate solution into plate and incubated plate for 30 min at 37 °C. After washing the plate for 5 times, added 100 µL chromogenic substrate into the plate to develop color at 37 °C for 15 min and then 50 µL stop solution was used to stop color development. The concentration of MMP13 was measured by examining the OD value at 450 nm.

The concentrations of CTX-I and PINP in the serum derived from MM patients or mice were measured by using CTX-I ELISA kit and PINP ELISA kit according to the manufacturer's instructions. Briefly, diluted serum with sample silution buffer by the ratio of 1: 1. Added 50 µL diluted serum into the plate, and then added 50 µL biotin-labeled CTX-I antibody or CTX-I antibody quickly, followed by incubation for 1 h at 37 °C. After incubation, removed liquid and then washed the plate with wash buffer for 3 times. Subsequently, added 100 µL HRP-conjugated streptavidin into the plate and incubated the plate for 30 min at 37 °C. After wash plate for 3 times with wash buffer, added 50 µL substrate A and 50 µL substrate B into the plate and incubated it in dark for 10 min at 37 °C. 50 µL stop solution was used to stop the reaction in the plate.

The concentrations of CTX-Iand PINP were measured by examining the OD value at 450 nm.

**Flow cytometry**. Apoptotic cells were labeled by FITC-conjugated Annexin V (US Everbright Inc, USA). Dead cells were labeled by PI (US Everbright Inc). Cell staining was performed according to the manufacturer's instructions. The apoptotic cells include the early apoptotic cells (Annexin V+, PI–) and late apoptotic cells (Annexin V+, PI+). Cells were then analyzed by CytExpert software (Beckman Coulter, USA), and the results were analyzed using FlowJo v10.0.7 software (BD Biosciences, CA). Each test was repeated three times.

**Statistical analysis**. All data are shown as means ± SD. Unpaired two-sided $t$-test was used to compare two experimental groups. ANOVA one-way test was used to compare three or more experimental groups. Survival curves were analyzed using Log-rank test. The correlations of glycine expression with clinical characteristics were measured using the chi-square test. Pearson correlation analysis was used to measure the correlation between two groups. Significance was set at $p < 0.05$. GraphPad Prism 7 or SPSS 20.0 was used for analysis.

**Reporting summary**. Further information on research design is available in the Nature Research Reporting Summary linked to this article.

## Data availability
The raw data of RNA-sequencing reported in this study has been deposited in the public database of Genome Sequence Archive (GSA) in National Genomics Data Center under the accession number HRA002453, which is accessible at https://ngdc.cncb.ac.cn/gsa-human/browse/HRA002453. The untargeted metabolomics data was included in an Excel form which is named as untargeted metabolomic data in the Source Data file. Source data are provided in Source Data file. Source data are provided with this paper.

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

## Acknowledgements

The authors thank professor Rushi Liu and Kaiqun Ren from Hunan Normal University School of Medicine for technical assistance. Thanks to the Animal Center of Hunan Normal University School of Medicine for providing the experimental platform. W.Z. is supported by the Ministry of Science and Technology of China under award number 2018YFA0107800, National Natural Science Foundation of China under award number 82130006, 81974010, Haihe Laboratory of Cell Ecosystem Innovation Fund under award number HH22KYZX0030, Natural Science Foundation of Hunan Province under award number 2020WK2006, Strategic Priority Research Program of Central South University under award number ZLXD2017004 and SKLEH-Pilot Research Grant under award number ZK22-06. J.X. is supported by Natural Science Foundation of Hunan Province under award number 2019JJ50838, PhD Scientific Research Start-up Fund of University of South China under award number 200XQD075, China Postdoctoral Science Foundation under award number 2018M640762 and Postdoctoral Science Foundation of Central South University under award number 198465. J.Z. is supported by Hunan Provincial Innovation Foundation for Postgraduates under award number CX20200261 and Fundamental Research Fund for Graduate of Central South University under award number 2020zzts221. Y.W. is supported by Fundamental Research Fund for Graduate of Central South University under award number 2020zzts783. We thank Dr. Daniel Ackerman (Insight Editing London) for editing the manuscript during submission.

## Author contributions

W.Z., W.J., and J.X. designed the research. J.X. and J.Z. performed the experiments and analyzed the data. J.Z., X.W., W.D., H.Y., X.L., G.A., L.Q., Q.Y., Y.S., and Q.L. collected clinical samples. W.D., Y.Z., Z.L., G.X., Q.W., X.F., J.G., B.M., Y.H., and X.J. provided technical assistance. W.Z., W.J., J.X., and J.Z. wrote and revised the manuscript. All authors read and approved the final manuscript.

## Competing interests

The authors declare no competing interests.
