## [Peer Review File · Nature Communications]

Title: Blocking glycine utilization inhibits multiple myeloma progression by disrupting glutathione balance and exhibits therapeutic potentialReviewers' comments:

Reviewer #1 (Remarks to the Author):

This is an interesting manuscript that identifies a role for glycine in the progression of myeloma. Following metabolomic analysis of patient samples, increased levels of glycine are found in myeloma patients and linked to disease indices. The authors clearly demonstrate a role for glycine in myeloma cell viability, mediated by downstream glutathione. While the studies on glycine utilization in myeloma are very thorough, the subsequent studies investigating targeting glycine in combination with bortezomib, and suggesting that collagen degradation is responsible for the elevated glycine are less robust. The title, abstract and discussion are heavily over concluded, for example, none of the studies begin to address the role of glycine in drug resistance yet this is in the title. The statement in the abstract "the elevation of glycine in the BM microenvironment was caused by the degradation of bone collagen mediated by MMP13" is not substantiated by the data. Overall, this manuscript has the potential to be of high impact, of interest to those studying myeloma, and other bone cancers, however substantial work is required to solidify the conclusions.

Major points

1. As mentioned above, overconcluding
2. For the experiments with end-point data and survival curves, when was the endpoint data done? It is not possible to compare tumour burden or bone disease from a survival curve experiment when animals were culled on different days reflecting disease severity
3. Were the mice treated with the glycine-free diet for just one week or for the entire experiment? What is the normal concentration of glycine in the diet
4. Why is a mouse model with Arp cells forming a 'tumour knot' used? This is not a well known model, not referenced, no methods are provided and it is not even clear from the images where these tumours are, as a standard xenograft model is on the flank. This does not seem to be a model of myeloma, it does not incorporate the bone microenvironment at all.
5. What effect does GSH have alone in vivo, this is an important control
6. Figure 1C shows an increase in glycine in newly diagnosed MM patients as compared to control, yet extended figure 5 shows a reduction in glycine in relapsed patients. This does not make sense and is not explained
7. The studies on resistance are limited. The only evidence for a link is between the SLC6A9 and bortezomib response, with a greater response to bortezomib when the glycine transporter is knocked down
8. The authors do not provide anywhere near enough information to state that 'glycine accumulation in the BM microenvironment was caused by MMP13-induced degradation of bone collagen'. Glycine is increased in patients with osteolysis, no indications of bone parameters or patient characteristics are provided, it is surprising that they have more patients without bone destruction given the nature of myeloma. It is important to investigate whether glycine levels correlate with markers of bone remodelling (CTX, P1NP, TRAP etc). Regarding the link with MMP13, inhibition of MMP13 is well known

to reduce tumour burden, which would be expected to have a consequent reduction in glycine levels, no causative link is demonstrated. Furthermore, there is no information on the MMP13 inhibitor CL-12898, I can't find this in the manuscript or in an internet search.

Minor points

- The authors do not accurately reflect the current knowledge in the field. No mention of the significant number of metabolomic studies already performed in myeloma patients is made, nor the fact that glutathione has already been linked to drug resistance in myeloma, Starheim et al. 2016, including by the authors themselves (Wu et al. 2020).
- There are a number of figures in the extended data (eg. Figure 2) with no stats
- Figures and figure legends need more detail, n needs to be included, are the error bars SEM or SD etc
- Far more details are needed in the methods, no information is given about any of the in vivo experiments, what was the diet, how was GSH administered etc.

At present, it would be impossible to reproduce large parts of this work, as insufficient detail is provided

Reviewer #2 (Remarks to the Author):

The article by Xia et al. investigates metabolite levels in the bone marrow microenvironment of multiple myeloma. This is a very interesting and timely topic since nutrient availability is known to affect metabolism and metabolic susceptibilities. This study extends this concept to MM and identifies a role for glycine. Glycine was found to be elevated in the BM microenvironment, a transporter was SLC6A9 was identified and the effect was rescued with betaine supplementation. And it was shown that the glycine cleavage system was required. They also provide some preliminary evidence that breakdown of bone collagen matrix is the cause of the elevated glycine levels. This part is interesting albeit speculative and the authors should temper these conclusions. Additionally the technical level of this study is performed to a high standard. Overall, this is an important study that will make an impact in the cancer metabolism and myeloma fields.

Reviewer #3 (Remarks to the Author):

The manuscript by Xia et al., describes an unexpected role for glycine in the bone microenvironment of multiple myeloma (MM) cells. It is well established that cancer cells in general show net excretion of (serine derived) glycine and that glycine starvation alone (in contrast to serine/glycine starvation) has generally no negative impact on cancer cells (this aspect should be addressed in the discussion of this paper). Because of serine catabolism and concomitant net excretion of glycine, it is also well established that glycine accumulates in the microenvironment (in different tumors). In contrast, cancer cells relying on exogenous (collagen derived) glycine is rather novel and unexpected. Thus, in principle, the here

presented findings are of interest and could highlight a specific metabolic phenotype of MM cells within the bone microenvironment. The authors demonstrate by different interventions (chemical and genetic +/- rescues) that extracellular glycine seems to be important for MM cells. What I do not understand is why cells break down glycine by GCS when the canonical function of glycine consumption is to provide glycine for GSH synthesis. To my understanding the breakdown of glycine would be a way to provide 1C units for the folate cycle, to generate additional NADH and/or to remove excessive glycine levels from the microenvironment which otherwise could become toxic.

While the principle finding (glycine consumption) is of interest, the interpretation is not very convincing. Most importantly, the overall presentation of the data is very weak and lacks scientific rigor:

- Throughout the manuscript, presented data do not contain information on number of n. As one example please see figure 3E. Also, it is not clear why in some panels (e.g. Figure 2F) several mice are shown while in figure 3E only one picture is shown. This varies throughout the manuscript. While it is OK to show one representative image, the figure (legend) should include the number of mice used.

- In general, it would help the reader to show single data points within each bar graph.

- The authors should make sure that all methods applied are properly explained within the method section.

- The 13C tracing approach (Figure 3B) raises several questions:

1. As glycine is a required building block for purines, why did labelled glycine did not label purines? Was the 6h time point sufficient to reach isotopic steady state?

2. The authors, write that GCS is reversible and see this as a possibility to obtain M+1 labeled GSH. However, in this case one should also expect M+1 glycine. Moreover, based on the cycling activity of the folate cycle (activity of SHMT), if glycine is oxidized by GCS to form 5,10-CH₂-THF, M+1 and M+3 serine should be the result. However, only M+2 serine was measured.

3. The cartoon in panel 3C is misleading. To my knowledge GCS derived 5,10CH₂-THF can not directly feed into GSH.

4. Did the authors correct for natural isotope abundance?

5. The method section is very poor. Only referring to a past publication is not sufficient.

- Applied statistical tests are not always correct (t-test instead of ANOVA)

- Control condition is often set to 1 (without any experimental error). However, statistical significance has been calculated on this basis. As only one example, I am wondering if the presented differences in figure 4E or 3D and E is really meaningful?

- Panel 3i lacks any information on colour code and fold change

- The data of the two presented cell lines in extFig3D looks extremely similar. Is it possible that there was a mixup?

- Figure 5L: why is there no increase in ROS levels with betaine alone (third bar). By blocking glycine consumption, it should also increase ROS levels (as shown in other panels). As betaine can by itself donate 1C units into the folate cycle it has the caveat that it can cause additional effects. An alternative approach would be helpful to consolidate the findings.

- ExtFig5F: in contrast to the text I do think that also glucose (esp at 5nM) has some rescue effect (as glycine). Moreover, statistical analysis is missing.

- Fig6C: Could it be possible that this data is biased by different tumor stage? I would assume that

patients with bone metastasis are in average in a more advanced stage compared to those without bone metastasis. As glycine extraction is a general phenomenon by cancer cells, increased plasma glycine levels could be a general phenotype of late stage cancer patients.

- Figure 6E: this correlation is not very convincing. Excluding the three outliers, it could even be a negative correlation.

- Figure 6H,K: I believe the correct name of the compound is CL82198 (not CL-12898)

Scientific comments:

- The authors show differences in intracellular glycine levels. However, the key readout to proof that cells indeed show net consumption of glycine would be to measure the uptake rate of glycine and by looking at glycine concentrations at the beginning and the end of the experiment. Also, it would support the findings when the authors demonstrate that the culture medium is not getting depleted of serine throughout the experiment. Because it is known that cells start consuming glycine once serine is depleted.

- Based on the growth arresting conditions (without necessary increase in cell death (extFig5F)) I am wondering if glycine deprivation might primarily affect purine de novo synthesis (as glycine is also an essential building block for purines). Similar as with antifolates, inhibition of nucleotide synthesis causes growth arrest without a strong increase in cell death. Therefore, to exclude this possibility it would be helpful to look at intermediates of the purine synthesis pathway (GAR, AICAR etc) and at total purine levels as well as to measure the flux into purines (e.g. by prolonging the ¹³C glycine experiment and/or by using a ¹³Cserine tracer).

- The authors can demonstrate good rescue efficiency by using GSH. However, GSH has a caveat as it can also be broken down into glycine and therefore (for example) also rescue purine synthesis in parallel. To corroborate that the rescue effect is really related to ROS, alternative ROS scavengers such as Trolox would be a good additional control.

- It is not clear to me why MM cells (in contrast to other cells) can not survive on serine alone. Thus, another possible explanation on why MM cells consume glycine could be that such cells have a defect in serine consumption. In this case, an alternative substrate could be glycine and formate to sustain purine and GSH synthesis. Did the authors check if the cells are able to consume serine? Related to this possibility: do the cells release or consume formate?

Dear Reviewers:

Thank you for your comments concerning our manuscript entitled "**Inhibition of glycine metabolism suppresses the progression and drug resistance of multiple myeloma**" (Full Paper, ID NCOMMS-21-05532-T). Those comments are all valuable and very helpful for us, both for revising and improving our paper and for guiding our research. We have studied the comments carefully and have made corrections which we hope will meet the publication standard. The revised portions are marked in red in the paper. The main corrections in the paper and the responses to the reviewers' comments are as follows:

Reviewer #1 (Remarks to the Author):

This is an interesting manuscript that identifies a role for glycine in the progression of myeloma. Following metabolomic analysis of patient samples, increased levels of glycine are found in myeloma patients and linked to disease indices. The authors clearly demonstrate a role for glycine in myeloma cell viability, mediated by downstream glutathione. on glycine utilization in myeloma and very thorough, the subsequent studies investigating targeting glycine in combination with bortezomib, and suggesting that collagen degradation is responsible for the elevated glycine are less robust.

The title, abstract and discussion are heavily over concluded, for example, none of the studies begin to address the role of glycine in drug resistance yet this is in the title. The statement in the abstract "the elevation of glycine in the BM microenvironment was caused by the degradation of bone collagen mediated by MMP13" is not substantiated by the data.

Overall, this manuscript has the potential to be of high impact, of interest to those studying myeloma, and other bone cancers, however substantial work is required to solidify the conclusions.

Major points

Comment 1: As mentioned above, overconcluding

Response to comment 1:

We appreciate your valuable comments. According to your suggestion, we revised our manuscript in two respects: firstly, to address the role of glycine in drug resistance, we examined whether glycine deprivation increased the sensitivity of MM cells to bortezomib (BTZ, the first line

treatment). We also examined the proliferation and apoptosis of MM cells with or without glycine transporter SLC6A9 knockdown after treatment with BTZ (Fig. R1a-c became Fig. 5a-c, new data; Fig. R1d became Fig. 6a, new data; Fig. R1e-g became Fig. 6e-f and Extended Data Fig. 6c; Fig. R1h became Extended Data Fig. 6e, new data). We then explored the functional roles of glycine in MM tumorigenesis and BTZ resistance in 5TGM1 cells and xenograft MM mouse models (Fig. R1i-l became Fig. 5d-f and Fig. 5h, new data; Fig. R1m became Extended Data Fig. 5e, new data). These findings indicated that glycine deprivation enhanced the effect of BTZ on MM by decreasing GSH levels in MM cells.

Secondly, based on our findings and conclusions, we edited our manuscript title to avoid overconcluding (from “Inhibition of glycine metabolism suppresses the progression and drug resistance of multiple myeloma” to “**Blocking glycine utilization inhibits progression of multiple myeloma by disrupting glutathione balance and exhibits therapeutic potential**”).

Figure R1. Glycine promotes drug resistance in MM.

a Glycine concentrations in ARP1 cells treated with different doses of BTZ (0, 2.5, or 5 nM) (mean ± sd, n = 3 for each group). **b** ARP1 cells cultured with or without glycine were treated with BTZ for 48 h; a CCK-8 assay was then performed to examine cell viability (mean ± sd, n = 3 for each group). **c** ARP1 cells cultured with (10 mg/L) or

without exogenous glycine were treated with different doses of BTZ (0, 2.5, 5, 10, or 20 nM) and with (100 μ M) or without Trolox for 48 h, followed by a CCK-8 assay (mean \pm sd, n = 3 for each group). **d** Relative mRNA expression levels of SLC6A5 and SLC6A9 in ARP1 cells treated with different doses of BTZ (0, 2.5, or 5 nM) (mean \pm sd, n = 3 for each group). **e** CCK-8 assay of ARP1 Scramble (Scr), ARP1 SLC6A9sh1, and ARP1 SLC6A9sh2 cells treated with different doses of BTZ (0, 2.5, 5, or 10 nM) (mean \pm sd, n = 3 for each group). **f** Clonogenic analysis of ARP1 Scr, ARP1 SLC6A9sh1, and ARP1 SLC6A9sh2 cells with (5 nM) or without BTZ treatment (mean \pm sd, n = 3 for each group). **g** The percentage of apoptotic cells among ARP1 Scr, ARP1 SLC6A9sh1, and ARP1 SLC6A9sh2 cells after BTZ treatment (mean \pm sd, n = 3 for each group). **h** ROS levels in ARP1 Scr and ARP1 SLC6A9sh1 cells treated with BTZ (5 nM) and with (100 μ M) or without Trolox (mean \pm sd, n = 3 for each group). **i** Live imaging of tumor-associated luminescence intensity in 5TGM1 MM mice in groups fed with control diet, fed with control diet (n=6) and treated with BTZ (1 mg/kg, 3 times/week) (n=6), fed with glycine-free diet (n=6), fed with glycine-free diet and treated with BTZ (1 mg/kg, 3 times/week) (n=6), or fed with glycine-free diet and treated with BTZ (1 mg/kg, 3 times/week) plus GSH (2 mg/kg, 3 times/week) (n=6). **j-k** The luminescence intensities and serum IgG2b concentrations of 5TGM1 MM mice in these five groups (mean \pm sd). **l** The survival curves of 5TGM1 MM mice in these five groups. **m** The xenograft MM model was prepared by injecting ARP1 cells (1×10^6 for each mouse) into the right abdomen of B-NDG mice. After one week, mice were divided into five groups: control diet (n=3), control diet + BTZ (n=3), glycine-free diet(n=3), glycine-free diet + BTZ(n=3), and glycine-free diet + BTZ + GSH (n=3). The tumor burden of B-NDG mice in these five groups. *p < 0.05, **p < 0.01, ***p < 0.001, ns: not significant (p > 0.05). p values were calculated using two-tailed unpaired Student's t-tests (a, b, c, d, e, g, h, j). p value was calculated using Log-rank test (l). p values were calculated using ANOVA test (k, m).

Comment 2: For the experiments with end-point data and survival curves, when was the endpoint data done? It is not possible to compare tumour burden or bone disease from a survival curve experiment when animals were culled on different days reflecting disease severity

Response to comment 2:

Based on the experimental animal welfare ethics of Central South University, all animals are observed daily for viability. Animals inoculated with tumors are observed and evaluated twice weekly to assess their in-depth physical condition and evaluate tumor burden. Mice are humanely euthanized when they become sick (symptoms include weight loss, appetite loss, hunched posture, leg lame, or bone fracture) and have high tumor burden.

In the present study, tumor burden was monitored by examining tumor-associated luminescence intensity and IgG2b concentration in all groups at the same timepoints. Bone disease is one of the main features of MM, and we thus also analyzed bone destruction in mice by Micro-CT scanning and examination of the bone remodeling markers CTX-1 and PN1P in all groups. Examination of CTX-1 and PN1P was performed at 4 weeks in all groups according to the reviewer's suggestion (Fig. 2k-l, Fig. 5g, Fig. 7k). Micro-CT scanning of mouse femurs was performed after they become sick (symptoms include weight loss, appetite loss, hunched posture, leg lame, or bone fracture) and have high tumor burden. Above all, we reflected disease severity by examining tumor burden, including luminescence intensity, serum IgG2b concentration, and Micro-CT, at the same timepoints in all groups.

Comment 3: Were the mice treated with the glycine-free diet for just one week or for the entire experiment? What is the normal concentration of glycine in the diet.

Response to comment 3:

Thank you for your question. We fed the mice with the glycine-free or control diet for the entire experiment to maintain the concentration of glycine. According to the formula of feed from the Trophic Animal Feed High-Tech Company (China), the concentration of glycine is 23 g/kg in the control diet. Meanwhile, glycine is depleted completely in the glycine-free diet, and the other non-essential amino acid levels are increased proportionally to achieve the same total amino acid content. The composition of the diet was included in Materials and Methods of the revised manuscript.

Comment 4: Why is a mouse model with Arp cells forming a 'tumour knot' used? This is not a well known model, not referenced, no methods are provided and it is not even clear from the images where these tumours are, as a standard xenograft model is on the flank. This does not seem to be a model of myeloma, it does not incorporate the bone microenvironment at all.

Response to comment 4:

Thank you for this comment. Human MM cells injected into a B-NDG mouse (NOD.CB17-PrkdcscidIl2rgtm1/Bcgen, Biocytogen Co, Beijing, China) and forming a 'tumour knot' is a xenograft mouse model that has been used previously in our^[1-2] and others' publications^[3-5]. The major advantages of this mouse model are the use of humanized MM cell lines and the feasibility

of gene editing. Therefore, in this study, we used the mouse model to detect tumorigenesis after ARP1 GLDC knockdown or in mice fed with/without the glycine-free diet. To further examine the bone microenvironment, we also incorporated the 5TGM1 MM mouse model in this study. In this model, inoculated MM cells travel to the bone marrow, allowing study of MM growth directly in the BM microenvironment^[6].

- [1] Zhou W. *et al.* NEK2 induces drug resistance mainly through activation of efflux drug pumps and is associated with poor prognosis in myeloma and other cancers. *Cancer Cell* **23**, 48-62 (2013).
- [2] Xia JL. *et al.* NEK2 induces autophagy-mediated bortezomib resistance by stabilizing Beclin-1 in multiple myeloma. *Molecular Oncology* **14**, 763-778 (2020).
- [3] Qin Y. *et al.* Epigenetic silencing of miR-137 induces drug resistance and chromosomal instability by targeting AURKA in multiple myeloma. *Leukemia* **31**, 1123–1135 (2017).
- [4] Yan P. *et al.* FOXO3-engineered human ESC-derived vascular cells promote vascular protection and regeneration. *Cell Stem Cell* **24**, 447-461(2019).
- [5] Zhang F. *et al.* Specific decrease in B-cell-derived extracellular vesicles enhances post-chemotherapeutic CD8(+) T cell responses. *Immunity* **50**, 738-750(2019).
- [6] Jian XX. *et al.* Alterations of gut microbiome accelerate multiple myeloma progression by increasing the relative abundances of nitrogen-recycling bacteria. *Microbiome* **8**, 74(2020)

Comment 5: What effect does GSH have alone in vivo, this is an important control

Response to comment 5:

Thank you very much for your constructive comments and suggestions. We reperformed this study using four groups of mice: control diet, control diet + GSH, glycine-free diet, and glycine-free diet + GSH (Figure R2a-d). The results, which were included in the revised manuscript (Fig. 3f-i), showed that GSH promotes MM progression (labeled in red in the revised manuscript).

Figure R2. GSH promotes MM progression.

a Live imaging luminescence intensity of 5TGM1 MM mice groups fed with control diet (n=6), fed with control diet and treated with GSH (2 mg/kg, 3 times/week) (n=6), fed with glycine-free diet (n=6), or fed with glycine-free diet and treated with GSH (2 mg/kg, 3 times/week) (n=6). **b-c** The luminescence intensities and serum IgG2b concentrations of these five groups of 5TGM1 MM mice (mean ± sd). **d** The survival curves of these five groups of 5TGM1 MM mice. *p < 0.05, **p < 0.01, ns: not significant (p > 0.05). p values were calculated using two-tailed unpaired Student's t-tests (b). p values were calculated using ANOVA test (c). p value was calculated using Log-rank test (d).

Comment 6: Figure 1C shows an increase in glycine in newly diagnosed MM patients as compared to control, yet extended figure 5 shows a reduction in glycine in relapsed patients. This does not make sense and is not explained.

Response to comment 6:

Thank you for your valuable comment. BTZ resistance is the main cause for relapse in MM. Our data indicated that exogenous glycine enters into MM cells, where it becomes involved in GSH synthesis. Exogenous glycine deprivation sensitizes MM cells to BTZ, while the addition of GSH reduces this sensitivity both *in vitro* and *in vivo*. Moreover, BTZ treatment can elevate the expression of glycine transporter SLC6A9 and thus increase glycine uptake (and thereby GSH levels) in MM cells (Fig. R3a became Fig. 6a, new data; Fig. R3b became Fig. 5a, new data; Fig. R3b c, new data). To assist in responding to this comment, we also measured glycine concentrations in primary MM

cells and bone marrow supernatants derived from five newly diagnosed and five relapsed MM patients. We found that relapsed MM patients had higher intracellular glycine (in MM cells) and lower bone marrow glycine than did newly diagnosed patients (Fig. R3d became Extend Data Fig. 5b, new data). These findings indicate that MM cells need to absorb glycine from the microenvironment to maintain drug resistance. We think that this is why glycine is reduced in the bone marrow microenvironment of relapsed patients.

Figure R3. BTZ-resistant MM cells need more glycine to maintain their proliferation and drug resistance.

a Expression of SLC6A5 and SLC6A9 in ARP1 cells treated with BTZ (0, 2.5, or 5 nM) for 24 h (mean ± sd, n = 3 for each group). **b** Intracellular glycine concentrations in ARP1 cells treated with different doses of BTZ (0, 2.5, or 5 nM) for 24 h (mean ± sd, n = 3 for each group). **c** GSH levels in ARP1 cells with (5 nM) or without BTZ treatment (mean ± sd, n = 3 for each group). **d** Glycine concentrations in the BM and CD138⁺ cells of newly diagnosed and relapsed MM patients (mean ± sd, n = 3 for each group). *p < 0.05, **p < 0.01, ns: not significant (p > 0.05). p values were calculated using two-tailed unpaired Student's t-tests (a, b, c, d).

Comment 7: The studies on resistance are limited. The only evidence for a link is between the SLC6A9 and bortezomib response, with a greater response to bortezomib when the glycine transporter is knocked down

Response to comment 7:

We appreciate your valuable comments and suggestion. We designed several experiments to support our conclusion that glycine promotes drug resistance in MM cells. These results were included in

Figure 5 in our revised manuscript. As shown in Figure 3 above, we found that BTZ treatment can increase the expression of glycine transporter SLC6A9 and thus increase glycine uptake, thereby elevating GSH levels in MM cells. Other groups have reported that GSH is effective in promoting BTZ resistance in MM cells. We thus speculate that glycine may contribute to BTZ resistance in MM cells by elevating GSH levels. Next, we found that glycine deprivation increases the effects of BTZ on MM, while the addition of GSH reverses the impact of glycine deprivation both *in vitro* and *in vivo* (Fig. R4a-f became Fig. 5a-f, new data; Fig. R4g became Fig. 5h, new data; Fig. R4h became Extended Data Fig. 5e). We also showed that blocking glycine utilization by SCL6A9 knockdown or treatment with betaine, a competitive analogue of glycine, significantly increases the effect of BTZ on MM (Figure 6 in the revised manuscript). These findings indicated that glycine utilization promotes BTZ resistance in MM.

Figure R4. Glycine deprivation sensitizes MM cells to BTZ.

a Intracellular glycine concentrations in ARP1 cells treated with different doses of BTZ (0, 2.5, or 5 nM) for 24 h (mean \pm sd, n = 3 for each group). **b** ARP1 cells cultured with (10 mg/L) or without exogenous glycine were treated with different doses of BTZ (0, 2.5, 5, 10, or 20 nM) for 48 h followed by a CCK-8 assay (mean \pm sd, n = 3 for each group). **c** ARP1 cells cultured with (10 mg/L) or without exogenous glycine were treated with different doses of BTZ (0, 2.5, 5, 10, or 20 nM) and with (100 μ M) or without Trolox for 48 h. This was followed by a CCK-8 assay (mean \pm sd, n = 3 for each group). **d** Live imaging luminescence intensity of 5TGM1 MM mice in groups fed with control diet (n=6), fed with control diet and treated with BTZ (1 mg/kg, 3 times/week) (n=6), fed with glycine-free diet (n=6), fed with glycine-free diet and treated with BTZ (1 mg/kg, 3 times/week) (n=6), or fed with glycine-free diet and treated with BTZ (1 mg/kg, 3times/week) plus GSH (2 mg/kg, 3 times/week) (n=6). **e-f** The luminescence intensities and serum IgG2b concentrations of 5TGM1 MM mice in these five groups (mean \pm sd). **g** The survival curves of these five groups of 5TGM1 MM mice. **h** The tumor burdens in B-NDG mice in these five groups (mean \pm sd, n = 3 for each group). *p < 0.05, **p < 0.01, ***p < 0.001, ns: not significant (p > 0.05). p values were calculated using two-tailed unpaired Student's t-tests (a, b, c, e). p value was calculated using Log-rank test (g). p values were calculated using ANOVA test (f, h).

[1] Wu X, *et al.* Phosphoglycerate dehydrogenase promotes proliferation and bortezomib resistance through increasing reduced glutathione synthesis in multiple myeloma. *Br J Haematol* **190**, 52-66 (2020)

[2] Starheim KK, *et al.* Intracellular glutathione determines bortezomib cytotoxicity in multiple myeloma cells. *Blood Cancer J* **6**, e446 (2016)

Comment 8: The authors do not provide anywhere near enough information to state that 'glycine accumulation in the BM microenvironment was caused by MMP13-induced degradation of bone collagen'. Glycine is increased in patients with osteolysis, no indications of bone parameters or patient characteristics are provided, it is surprising that they have more patients without bone destruction given the nature of myeloma. It is important to investigate whether glycine levels correlate with markers of bone remodelling (CTX, P1NP, TRAP etc). Regarding the link with MMP13, inhibition of MMP13 is well known to reduce tumour burden, which would be expected to have a consequent reduction in glycine levels, no causative link is demonstrated. Furthermore, there is no information on the MMP13 inhibitor CL-12898, I can't find this in the manuscript or in an internet search.

Response to comment 8:

Thank you for your constructive suggestions. In the present study, bone disease of MM patients was examined by CT. We divided MM patients into two classes based on extent of bone disease: patients with osteoporosis but without bone destruction (considered “slight bone disease”) and patients with obvious bone destruction (considered “serious bone disease”). The detailed clinical information of bone disease in MM patients is listed in Supplementary Table S6. We analyzed the correlation between glycine levels and bone destruction in these MM patients, finding that patients with bone destruction have higher serum glycine levels than do those without bone destruction (Fig. R5a became Fig. 7c). We also detected the two markers of bone remodeling (CTX-1 and P1NP) in patients’ serum and correlated these with glycine levels. We found that CTX-1 and P1NP levels were both positively correlated with glycine levels (Fig. R5b became Fig. 7d, new data). These data suggested that serum glycine levels correlate with bone remodeling.

Our untargeted metabolic analysis showed that the main amino acid components of bone collagen, including glycine, glutamic acid, proline, and alanine were increased in the bone marrow of MM patients compared to healthy donors. Bone collagen is degraded by matrix metalloproteinases (MMPs), and bone destruction is accompanied by degradation of bone collagen. Previous publications indicated that MMP13 secreted by MM cells promotes bone destruction in MM^[1-3]. In our study, we found that MMP13 was overexpressed in MM cells; moreover, serum MMP13 positively correlated with serum glycine levels and bone destruction in MM patients (Fig. R5c became Extended Data Fig. 7d; Fig. R5d became Fig. 7e, new data). We thus speculated that the increase of glycine in the bone marrow microenvironment may be caused by MMP13-induced bone destruction. Consistent with previous reports, we found that inhibition of MMP13 decreases bone destruction in 5TGM1 MM mice; meanwhile, glycine levels was also reduced in 5TGM1 MM mice treated with the MMP13 inhibitor CL-82198 (Fig. R5e-h became Fig. 7i-l). In view of the above data, we concluded that MMP13-induced bone destruction increases glycine levels in MM. To assist with satisfying this comment, we have also included information on CL-82198 in the Supplemental Materials and Methods section in red.

Figure R5. Bone destruction causes glycine accumulation in the MM bone marrow microenvironment.

a Glycine concentrations in serum derived from MM patients with or without bone destruction (mean \pm sd, without bone destruction: n = 24, bone destruction: n = 12). **b** The correlations between serum glycine and serum CTX-1 or P1NP in MM patients (n = 26). **c** MMP13 concentrations in serum derived from MM patients with or without bone destruction (mean \pm sd, without bone destruction: n = 24, bone destruction: n = 12). **d** The correlation between serum MMP13 concentration and serum glycine concentration in MM patients (n = 87) **e** Glycine concentrations in serum derived from 5TGM1 MM mice (mean \pm sd, n = 5 in each group). **f** The concentrations of CTX-1 and P1NP in serum derived from 5TGM1 MM mice (mean \pm sd, n = 5 in each group). **g** Micro-CT images of femurs derived from 5TGM1 MM mice treated with physiological saline or CL-82198 (2 mg / kg). **h** Quantification of bone microstructural parameters including trabecular bone volume fraction (BV / TV), trabecular number (Tb. N), trabecular separation (Tb. Sp), and trabecular thickness (Tb. Th) (mean \pm sd, n = 5 in each group). *p < 0.05, ns: not significant (p > 0.05). p values were calculated using two-tailed unpaired Student's t-tests (a, c, e, f, h).

[1] Tauro M. *et al.* Bone-seeking matrix metalloproteinase-2 inhibitors prevent bone metastatic breast cancer growth. *Mol Cancer Ther* **16**, 494-505 (2017).

[2] Jabłońska-Trypuć A. *et al.* Matrix metalloproteinases (MMPs), the main extracellular matrix (ECM) enzymes in collagen degradation, as a target for anticancer drugs. *J Enzyme Inhib Med Chem* **31**, 177-

183 (2016).

[3] Fu J. *et al.* Multiple myeloma-derived MMP-13 mediates osteoclast fusogenesis and osteolytic disease. *J Clin Invest* **126**, 1759-1772 (2016).

Minor points

Comment 9: The authors do not accurately reflect the current knowledge in the field. No mention of the significant number of metabolomic studies already performed in myeloma patients is made, nor the fact that glutathione has already been linked to drug resistance in myeloma, Starheim et al. 2016, including by the authors themselves (Wu et al. 2020).

Response to comment 9:

Thank you for this valuable information. We have included this knowledge in the introduction. The relevant references were included in the first paragraph of Discussion, and these references were highlighted in red in our revised manuscript.

Comment 10: There are a number of figures in the extended data (eg. Figure 2) with no stats.

Response to comment 10:

Thank you for pointing it out. We have shown the statistics of our data in the revised manuscript.

Comment 11: Figures and figure legends need more detail, n needs to be included, are the error bars SEM or SD etc.

Response to comment 11:

Thank you for pointing it out. We have added more detail to our figures and figure legends, including n and SD error bars, in the revised manuscript.

Comment 12: Far more details are needed in the methods, no information is given about any of the in vivo experiments, what was the diet, how was GSH administered etc.

Response to comment 12:

Thank you for pointing it out. We have improved the Materials and Methods section in the revised manuscript. The detailed experimental methods for the untargeted and targeted metabolomics assays, isotope tracing, micro-CT scanning, and mice experiments were included in the methods section,

and the others were included in the supplemental methods.

Specifically, the compositions of the control and glycine-free diets were included in the methods section of our revised manuscript, as follows: “The control diet was composed of essential and non-essential amino acids (including 23 g/kg of glycine), corn starch, dextrin and sucrose, cellulose, soybean oil, minerals, vitamins, choline chloride, and tert-butylhydroquinone. The glycine-free diet was the same as the control diet, but without glycine, and the other non-essential amino acid levels were increased proportionally to achieve the same total amino acid content.” Meanwhile, the BTZ (1 mg/kg), GSH (2 mg/kg) and betaine (500 mg/kg) was injected intraperitoneally according to the weight of the mice until the end of experiment.

The method for analyzing cellular GSH concentration was included in the supplemental methods as follows: “ 1×10^7 MM cells were collected via centrifugation (200 g for 5 min at 4 °C). Cells were washed twice with PBS followed by removal of the supernatant and resuspension in 80 μ L of 10 mM HCl. Cells were lysed via two freeze/thaw cycles followed by addition of 20 μ L of 5% SSA and centrifugation at 8,000 g for 10 min at 4 °C. The supernatant was moved to a new tube. SSA was diluted to 0.5% with ddH₂O. Finally, the intracellular GSH concentration was measured using a glutathione disulphide (GSSG)/GSH quantification kit II (Dojindo, Kumamoto, Japan) according to the manufacturer’s instructions.”

Comment 13: At present, it would be impossible to reproduce large parts of this work, as insufficient detail is provided

Response to comment 13:

Thank you for your comments and suggestions. In the revised manuscript, we have included additional details and explanatory text in the Materials and Methods and in the supplemental methods that we believe will provide researchers with the information necessary to reproduce our experiments.

Reviewer #2 (Remarks to the Author):

The article by Xia et al. investigates metabolite levels in the bone marrow microenvironment of multiple myeloma. This is a very interesting and timely topic since nutrient availability is known to affect metabolism and metabolic susceptibilities. This study extends this concept to MM and identifies a role for glycine. Glycine was found to be elevated in the BM microenvironment, a transporter was SLC6A9 was identified and the effect was rescued with betaine supplementation. And it was shown that the glycine cleavage system was required. They also provide some preliminary evidence that breakdown of bone collagen matrix is the cause of the elevated glycine levels. This part is interesting albeit speculative and the authors should temper these conclusions. Additionally the technical level of this study is performed to a high standard. Overall, this is an important study that will make an impact in the cancer metabolism and myeloma fields.

Response to comment:

Thank you for your positive comments.

Reviewer #3 (Remarks to the Author):

The manuscript by Xia et al., describes an unexpected role for glycine in the bone microenvironment of multiple myeloma (MM) cells. It is well established that cancer cells in general show net excretion of (serine derived) glycine and that glycine starvation alone (in contrast to serine/glycine starvation) has generally no negative impact on cancer cells (this aspect should be addressed in the discussion of this paper). Because of serine catabolism and concomitant net excretion of glycine, it is also well established that glycine accumulates in the microenvironment (in different tumors). In contrast, cancer cells relying on exogenous (collagen derived) glycine is rather novel and unexpected. Thus, in principle, the here presented findings are of interest and could highlight a specific metabolic phenotype of MM cells within the bone microenvironment. The authors demonstrate by different interventions (chemical and genetic +/- rescues) that extracellular glycine seems to be important for MM cells. What I do not understand is why cells break down glycine by GCS when the canonical function of glycine consumption is to provide glycine for GSH synthesis. To my understanding the breakdown of glycine would be a way to provide 1C units for the folate cycle, to generate additional NADH and/or to remove excessive glycine levels from the microenvironment which otherwise could become toxic.

While the principle finding (glycine consumption) is of interest, the interpretation is not very convincing. Most importantly, the overall presentation of the data is very weak and lacks scientific rigor:

Comment 1: Throughout the manuscript, presented data do not contain information on number of n. As one example please see figure 3E. Also, it is not clear why in some panels (e.g. Figure 2F) several mice are shown while in figure 3E only one picture is shown. This varies throughout the manuscript. While it is OK to show one representative image, the figure (legend) should include the number of mice used.

Response to comment 1:

Thank you for your constructive suggestion. We have included the numbers of mice used in each experiment in the figure legends of the revised manuscript.

Comment 2: In general, it would help the reader to show single data points within each bar graph.

Response to comment 2:

Thank you for this suggestion. As you recommended, we have added the single data points within each bar graph in the revised manuscript.

Comment 3: The authors should make sure that all methods applied are properly explained within the method section.

Response to comment 3:

We appreciate this constructive suggestion. We have improved the materials and methods section as described above by including detailed information on the untargeted and targeted metabolomics assays, isotope tracing, micro-CT scanning, and mouse models of MM. Additional details were included in the supplemental methods.

Comment 4: The ^{13}C tracing approach (Figure 3B) raises several questions: As glycine is a required building block for purines, why did labelled glycine did not label purines? Was the 6h time point sufficient to reach isotopic steady state?

Response to comment 4:

Thank you for this comment. We have redesigned the glycine metabolism flux experiment in the revised manuscript, increasing the number of cells detected and extending the timepoints to 18 hours as described in Jain *et al.* (2012)^[1]. The ^{13}C -labeled glycine and glycine metabolites, including serine and glutamic acid, cysteine, GSH, homocysteine, AMP, GMP, IMP, adenine, and xanthine were measured at 0, 2, 6, and 18 hours. As shown in Figure 3b and Extended Data Figure 3 in our revised manuscript, we obtained ^{13}C -labeled glycine, serine, GSH, purine metabolites AMP, GMP, IMP, adenine, and xanthine in ARP1 cells cultured with ^{13}C -labeled glycine (Fig. R6a became Fig. 3a, new data; Fig. R6b became Extended Data Fig. 3a, new data; Fig. R6c became Fig. 3b, new data; Fig. R6d).

Figure R6. Results from updated glycine metabolic flux experiments.

a Schematic of the updated glycine metabolic flux experiments. **b** The proportions of ^{13}C -labeled metabolites in ARP1 cells exposed to $^{13}\text{C}_2$ -glycine for 18 h (mean \pm sd, n = 3 in each group). **c** Schematic of glycine metabolism in MM cells. **d** The proportions of ^{13}C -labeled adenine and xanthine in ARP1 cells exposed to $^{13}\text{C}_2$ -glycine for 0, 2, 6, and 18 h (mean \pm sd, n = 3 in each group).

[1] Jain M. *et al.* Metabolite profiling identifies a key role for glycine in rapid cancer cell proliferation. *Science* **334**, 1040-1044 (2012).

Comment 5: The authors, write that GCS is reversible and see this as a possibility to obtain M+1 labeled GSH. However, in this case one should also expect M+1 glycine. Moreover, based on the cycling activity of the folate cycle (activity of SHMT), if glycine is oxidized by GCS to form 5,10-CH₂-THF, M+1 and M+3 serine should be the result. However, only M+2 serine was measured.

Response to comment 5:

Thank you for this insightful comment. In the reworked metabolic flux experiment, we obtained

M+1 labeled glutathione, M+2 labeled glycine, and M+2 labeled serine, but not M+1 labeled glycine or M+1/3 labeled serine (Fig. R7a became Extended Data Fig. 3a). However, we found that glycine deprivation decreased the expression of GLDC (Fig. R7b became Fig. 4c), a key enzyme of the glycine cleavage system, and the ratio of NADH/NAD⁺, which is upregulated by glycine cleavage (Fig. R7c became Fig. 4b; Fig. R7d became Fig. 4a). Moreover, GLDC knockdown significantly decreased the level of GSH, the ratio of NADH/NAD⁺, and MM cell proliferation both *in vitro* and *in vivo* (Fig. R7e-f became Fig. 4f-g; Fig. R7g became Fig. 4e; Fig. R7h-i became Fig. 4h-I; Fig. R7j-l became Fig. 4k-m). Taken together, these findings indicated that glycine is broken down by the glycine cleavage system in MM cells via the following reaction: Glycine + H₄folate + NAD⁺ ⇌ 5,10-methylene-H₄folate + CO₂ + NH₃ + NADH + H⁺. Since this reaction is reversible, isotope-labeled 5,10-methylene-H₄folate can react with unlabeled CO₂ to generate M+1 labeled glycine in MM cells. We thus concluded that isotope-labeled carbon in glutathione (M+1) was derived from M+1 labeled glycine. In our metabolic flux experiment, most M+1 labeled glycine was involved in glutathione synthesis, and therefore free M+1 labeled glycine was too low to be detected. We think that this is the reason why M+1 labeled glycine and M+1/3 labeled serine could not be detected in our metabolic flux experiment.

Figure R7. Glycine is broken down by the glycine cleavage system in MM cells.

a The proportions of ¹³C-labeled metabolites in ARP1 cells exposed to ¹³C₂-glycine for 18 h (mean ± sd, n = 3 in each group). **b** Western blots of GLDC, AMT, DLD, GCSH, and β-actin in ARP1 cells cultured with or without exogenous glycine for 24 h. **c** The ratios of NADH/NAD⁺ in ARP1 cells cultured with or without exogenous glycine for 24 h (mean ± sd, n = 3 in each group). **d** Schematic of the glycine cleavage system. **e** GSH levels in ARP1 scramble (Scr) and ARP1 GLDCsh1 cells (mean ± sd, n = 3 in each group). **f** The ratios of NADH/NAD⁺ in ARP1 Scr and ARP1 GLDCsh1 cells (mean ± sd, n = 3 in each group). **g** Numbers of ARP1 Scr, ARP1 GLDCsh1, and ARP1 GLDCsh2 cells after culturing for 0-6 days (mean ± sd, n = 3 in each group). **h** Numbers of ARP1 Scr, ARP1

GLDCsh1, and ARP1 GLDCsh1 cells treated with GSH (2 mg/L) after culturing for 0-6 days (mean \pm sd, n = 3 in each group). **i** Clonogenic analysis of ARP1 Scr, ARP1 GLDCsh1, and ARP1 GLDCsh1 cells treated with GSH (10 μ M) (mean \pm sd, n = 3 in each group). **j** Tumor knots developed in B-NDG mice injected with ARP1 Scr (n = 3) or ARP1 GLDCsh1 (n = 3) cells, and also from mice transplanted with ARP1 GLDCsh1 cells and treated with GSH (2 mg/kg) (n = 3). **k** Analysis of tumor volumes in ARP1 Scr, ARP1 GLDCsh1, and ARP1 GLDCsh2 mice (mean \pm sd, n = 3 in each group). **l** GSH levels in tumor knots derived from mice transplanted with ARP1 Scr or ARP1 GLDCsh1 cells with or without GSH treatment (mean \pm sd, n = 3 in each group). *p < 0.05, **p < 0.01, ***p < 0.001, ns: not significant (p > 0.05). p values were calculated using two-tailed unpaired Student's t-test (c, e, f, l). p values were calculated using ANOVA test (g, h, k).

Comment 6: The cartoon in panel 3C is misleading. To my knowledge GCS derived 5,10CH₂-THF can not directly feed into GSH.

Response to comment 6:

Thank you for this comment. We corrected the cartoon in panel 3C based on the recent results of our updated metabolic flux experiment (Fig. R8 became Fig. 3b). In our metabolic flux experiment, we detected ¹³C-labeled metabolites including glycine, serine, cysteine, glutamic acid, homocysteine, glutathione, and purines. Glutathione consists of glycine, glutamic acid and cysteine. We detected glutathione containing one and two ¹³C, but ¹³C-labeled glutamic acid, cysteine and homocysteine were not detected, indicating that the ¹³C in glutathione was derived from glycine. The existence of glutathione containing one ¹³C suggested that glycine was cleaved in MM cells. As discussed above, ¹³C₂-glycine is broken down into isotope-labeled 5,10-methylene-H₄folate, isotope-labeled CO₂, NH₃, and NADH; meanwhile, some isotope-labeled 5,10-methylene-H₄folate can react with unlabeled CO₂ to generate glycine containing one ¹³C. Glycine containing one ¹³C is involved in glutathione synthesis, which is likely the reason for the generation of glutathione containing one ¹³C.

Figure R8. Schematic of glycine metabolism in MM cells.

Comment 7: Did the authors correct for natural isotope abundance?

Response to comment 7:

Thank you for your question. In our metabolic flux experiment, ARP1 cells cultured with $^{13}\text{C}_2$ -glycine were collected at 0, 2, 6, and 18 h. Cells collected at 0 h were used as a reference for analysis. At the beginning, the abundance of ^{13}C -labeled metabolite was corrected by subtracting natural ^{13}C -labeled metabolites. The formula was as follows: $A_{\text{labeled metabolite after correction}} = A_{\text{labeled metabolite}} - A_{\text{un-labeled metabolite}} \times P_{\text{natural labeled metabolite}}$. $A_{\text{labeled metabolite after correction}}$ is the abundance of labeled metabolite after correction. $A_{\text{labeled metabolite}}$ is the abundance of labeled metabolite before correction. $A_{\text{un-labeled metabolite}}$ is the abundance of un-labeled metabolite. $P_{\text{natural labeled metabolite}}$ is the proportion of natural labeled metabolite to un-labeled metabolite in cells collected at 0 h. Finally, the proportion of corrected metabolite containing ^{13}C to total metabolite was calculated by using this formula: $P_{\text{labeled metabolite after correction}} = A_{\text{labeled metabolite after correction}} / A_{\text{total metabolite}}$. This information has been included in the Materials and Methods section in the revised manuscript.

Comment 8: The method section is very poor. Only referring to a past publication is not sufficient.

Response to comment 8:

Thank you for this comment. We have greatly increased the level of detail in our methods sections in the revised manuscript. Metabolomics assays, isotope tracing, micro-CT scanning, and *in vivo* experiments are discussed in the Materials and Methods, while additional details are included in the supplemental methods. The new content was highlighted in red in our revised manuscript.

Comment 9: Applied statistical tests are not always correct (t-test instead of ANOVA)

Response to comment 9:

Thank you very much for your comment. We have corrected the statistical methods in the revised manuscript according to the characteristics of data. In our revised manuscript, Student's t-test was used to compare two experimental groups, and ANOVA test was used to analyze data containing three or more experimental groups. Survival curves were analyzed by using Log-rank test. The correlations of glycine expression with clinical characteristics were measured using the Chi-square test. The descriptions of statistics were highlighted in red in the Figure legends of revised manuscript.

Comment 10: Control condition is often set to 1 (without any experimental error). However, statistical significance has been calculated on this basis. As only one example, I am wondering if the presented differences in figure 4E or 3D and E is really meaningful?

Response to comment 10:

Thank you for your constructive suggestions. Our statistical results were based on original raw values rather than relative values. Meanwhile, Student's t-tests confirmed that the relationships shown in Figure R9 (original figure 4E, 3D and E) were statistically meaningful (Fig. R9a became Fig. 3d and Extended Data Fig. 3d; Fig. R9b became Extended Data Fig. 3e; Fig. R9c became Fig. 3e; Fig. R9d became Fig. 6c; Fig. R9e became Fig. 6i). In the revised manuscript, these figures were improved by using the raw values instead of the relative values.

Figure R9. Revised figures using raw instead of relative values.

a GSH levels in ARP1 or 5TGM1 cells cultured with different concentrations of exogenous glycine (0, 2, or 10 mg/L) (mean \pm sd, n = 3 in each group). **b** ROS levels in ARP1 or 5TGM1 cells cultured with different concentrations of exogenous glycine (0, 2, or 10 mg/L) (mean \pm sd, n = 3 in each group). **c** GSH levels in tumor knots derived from mice fed with control diet or glycine-free diet (mean \pm sd, n = 3 in each group). **d** GSH levels in ARP1 Scramble (Scr), ARP1 GLDCsh1, and ARP1 GLDCsh2 mice (mean \pm sd, n = 3 in each group). **e** GSH levels in ARP1 or 5TGM1 cells treated with or without betaine (2 mg/L) (mean \pm sd, n = 3 in each group). *p < 0.05, **p < 0.01, ***p < 0.001, ns: not significant (p > 0.05). p values were calculated using two-tailed unpaired Student's t-tests (a, c, b, c, d, e).

Comment 11: Panel 3i lacks any information on colour code and fold change

Response to comment 11: Thank you very much for this comment. We have added information on color code and fold change in the revised Figure 3j (shown below as Figure R10).

Figure R10. The expression profile of glycine metabolism-related genes in ARP1 cells cultured with or without glycine (10 mg/L).

Comment 12: The data of the two presented cell lines in extFig3D looks extremely similar. Is it possible that there was a mixup?

Response to comment 12:

Thank you for your comment. We originally set the control condition to 1 in both cell lines' datasets, causing the graphs from the two presented cell lines to look similar. We now show the raw values rather than the relative values (see Fig. R11 below and Extended Data Fig. 3e) in the revised manuscript.

Figure R11. ROS levels in ARP1 and 5TGM1 cells cultured with different concentrations of exogenous glycine (0, 2, or 10 mg/L). * $p < 0.05$, ns: not significant ($p > 0.05$). p values were calculated using two-tailed unpaired Student's t -tests.

Comment 13: Figure 5L: why is there no increase in ROS levels with betaine alone (third bar). By blocking glycine consumption, it should also increase ROS levels (as shown in other panels). As betaine can by itself donate 1C units into the folate cycle it has the caveat that it can cause additional effects. An alternative approach would be helpful to consolidate the findings.

Response to comment 13:

We indeed found that treating myeloma cells with betaine alone significantly increased intracellular ROS levels ($p=0.019$). The original image was truncated, so there did not appear to be a significant difference from the control group. In the revised manuscript, we adjusted the scale of this graph to make the difference visually clearer (see Fig. R12 below and Fig.6o).

Figure R12. ROS levels in ARP1 cells treated with BTZ and betaine alone or in combination. * $p < 0.05$, **** $p < 0.0001$, ns: not significant ($p > 0.05$). p values were calculated using two-tailed unpaired Student's t -tests.

Comment 14: ExtFig5F: in contrast to the text I do think that also glucose (esp at 5nM) has some rescue effect (as glycine). Moreover, statistical analysis is missing.

Response to comment 14:

Thank you very much for your critical comments. We re-examined our results, and glucose seems to have a small rescuing effect. However, this effect was not statistically significant ($p=0.4$). We included this statistical analysis in Extended Data Figure 5f (now 5d, shown below as Fig. R13).

Figure R13. Addition of glycine enhanced the resistance of ARP1 cells to treatment with BTZ. $**p < 0.01$, $***p < 0.001$, ns: not significant ($p > 0.05$). p values were calculated using two-tailed unpaired Student's t-tests.

Comment 15: Fig6C: Could it be possible that this data is biased by different tumor stage? I would assume that patients with bone metastasis are in average in a more advanced stage compared to those without bone metastasis. As glycine extraction is a general phenomenon by cancer cells, increased plasma glycine levels could be a general phenotype of late stage cancer patients.

Response to comment 15:

Thank you very much for your question and insightful comments. According to the NCCN (National Comprehensive Cancer Network) guidelines, there are two clinical staging systems for MM: ISS (International Staging System) and DS (Durie-Salmon). Bone destruction is one of the indicators of DS stage, and our data on MM patients' clinical characteristics suggested that serum glycine levels increased with tumor progression (Fig. R14a-b). However, we also observed that serum glycine levels increased significantly with the severity of bone destruction in patients at the same DS Stage (Fig. R14c became Extended Data Fig. 7a), suggesting that glycine concentration is linked to both clinical stage and bone disease severity.

Figure R14. Serum glycine levels are correlated with bone destruction.

a Serum glycine concentrations in MM patients at different ISS stages (mean \pm sd, n = 21 in MM patients at ISS stage I, n = 30 in MM patients at ISS stage II, n = 58 in MM patients at ISS stage III). **b** Serum glycine concentrations in MM patients at different DS stages (mean \pm sd, n = 4 in MM patients at DS stage I, n = 8 in MM patients at DS stage II, n = 98 in MM patients at DS stage III). **c** Serum glycine concentrations in MM patients at DS Stage III with or without bone destruction (mean \pm sd, with bone destruction, n=12; without bone destruction, n=18). *p < 0.05, ns: not significant (p > 0.05). p values were calculated using two-tailed unpaired Student's t-tests (a, b, c).

Comment 16: Figure 6E: this correlation is not very convincing. Excluding the three outliers, it could even be a negative correlation.

Response to comment 16:

Thank you for your constructive suggestions. We expanded the number of patients from 38 to 87 and found that the correlation between glycine and MMP13 was positive (p=0.0038) (Fig. R15 became Fig. 7e).

Figure R15. Glycine concentration is positively correlated with MMP13 concentration in MM patients.

Comment 17: Figure 6H,K: I believe the correct name of the compound is CL82198 (not CL-12898)

Response to comment 17:

Thank you for pointing it out. We apologize for our error and have corrected it in the revised manuscript. The information of CL82198 was included and highlighted in red in the Supplemental methods of our revised Supplemental data.

Scientific comments:

Comment 18: The authors show differences in intracellular glycine levels. However, the key readout to prove that cells indeed show net consumption of glycine would be to measure the uptake rate of glycine and by looking at glycine concentrations at the beginning and the end of the experiment. Also, it would support the findings when the authors demonstrate that the culture medium is not getting depleted of serine throughout the experiment. Because it is known that cells start consuming glycine once serine is depleted.

Response to comment 18:

We greatly appreciate your suggestion. To address this comment, we cultured ARP1 cells in complete medium. Both cells and conditioned medium were collected at 2, 4, and 6 hours, and glycine concentrations in cells and conditioned medium were then detected by HPLC. As shown in Fig. R16a-b, the glycine concentrations in the medium were reduced at later timepoints. Meanwhile, the intracellular glycine concentrations were significantly elevated. In addition, we cultured ARP1 cells in medium supplemented with or without glycine for 2, 4, and 6 hours, followed by detection of intracellular glycine. We found that ARP1 cells cultured with glycine had significantly higher intracellular glycine concentrations than those cultured without glycine at all timepoints (Fig. R16c). These findings supported our conclusion that MM cells absorb glycine from the microenvironment and that exogenous glycine depletion decreases intracellular glycine concentrations in MM cells.

Figure R16. ARP1 cells absorb glycine from media.

a Glycine concentrations in conditioned medium extracted from ARP1 cells after culturing for 2, 4, or 6 days (mean \pm sd, n = 3 in each group). **b** Glycine concentrations in ARP1 cells after culturing for 2, 4, or 6 days (mean \pm sd, n = 3 in each group). **c** Glycine concentrations in ARP1 cells cultured with or without exogenous glycine for 2, 4, or 6 days (mean \pm sd, n = 3 in each group). *p < 0.05, **p < 0.01, ***p < 0.001. p values were calculated using two-tailed unpaired Student's t-tests (a, b, c).

Comment 19: Based on the growth arresting conditions (without necessary increase in cell death (extFig5F)) I am wondering if glycine deprivation might primarily affect purine de novo synthesis (as glycine is also an essential building block for purines). Similar as with antifolates, inhibition of nucleotide synthesis causes growth arrest without a strong increase in cell death. Therefore, to exclude this possibility it would be helpful to look at intermediates of the purine synthesis pathway (GAR, AICAR etc) and at total purine levels as well as to measure the flux into purines (e.g. by prolonging the ^{13}C glycine experiment and/or by using a ^{13}C serine tracer).

Response to comment 19:

Thank you for your constructive suggestions. We redesigned and performed the glycine metabolic flux experiment according to your suggestions. ARP1 cells were cultured with glycine-free medium supplemented with 5% DFBS and ^{13}C -labeled glycine (10 mg/L) for 0, 2, 6, and 18 hours. In addition to ^{13}C -labeled GSH, glycine, and serine, we also obtained ^{13}C -labeled purines that included IMP, GMP, AMP, xanthine, and adenine (Fig. R17a became Extended Data Fig. 3a). MM cells are prone to produce large amounts of immunoglobulins, causing an endoplasmic reticulum stress and the production of high levels of intracellular ROS^[1, 2]. To maintain intracellular redox homeostasis, it is required for MM cells to produce amount of GSH to against high levels of ROS. We found that MM cells have higher levels of intracellular glycine and GSH than normal B cells (Fig. R17b). Addition of GSH, but not purines, could significantly recover cell proliferation in ARP1 cells cultured without exogenous glycine (Fig. R17c became Extended Data Fig. 3c). We also found that GSH decreased glycine deprivation-induced DNA damage in ARP1 cells; moreover, GSH could reverse the impact of glycine deprivation on cell proliferation and BTZ efficacy both *in vitro* and *in vivo* (Fig. R17d became Fig. 3m; Fig. R17e became Fig.3c; Fig. R17f-h became Fig. 5e-f). These data showed that exogenous glycine contributes to cell proliferation and drug resistance in ARP1 cells by promoting GSH synthesis.

Figure R17. GSH, but not purines, promotes cell proliferation and BTZ resistance in MM cells.

a The proportions of ¹³C-labeled metabolites in ARP1 cells exposed to ¹³C₂-glycine for 18 h (mean ± sd, n = 3 in each group). **b** The concentration of glycine and GSH in GM12878, ARP1 and 5TGM1. **c** CCK-8 assay in ARP1 cells cultured with or without glycine (10 mg/L), GSH (10 μM), and purine (5 nM) (mean ± sd, n = 3 in each group). **d** Western blots of γ-H2AX, p-ATR, p-ATM, p-CHK1, p-CHK2, CDC25A, and β-actin in ARP1 cells cultured with or without glycine and/or with GSH treatment (10 μM) for 48 h. **e** Growth curve of ARP1 cells treated with or without GSH in glycine-free medium (mean ± sd, n = 3 in each group). **f** Live imaging of luminescence intensity in 5TGM1 MM mice. **g-h** The luminescence intensities and serum IgG2b concentrations of 5TGM1 MM mice in the

groups fed with the control diet, fed with control diet and treated with BTZ (1 mg/kg, 3 times/week), fed with glycine-free diet, fed with glycine-free diet treated with BTZ (1 mg/kg, 3 times/week), or fed with glycine-free diet and treated with BTZ (1 mg/kg, 3times/week) plus GSH (2 mg/kg, 3 times/week) (mean ± sd). *p < 0.05, **p < 0.01, ***p < 0.001, ns: not significant (p > 0.05). p values were calculated using two-tailed unpaired Student's t-tests (b, d, f). p values were calculated using ANOVA test (g).

[1] Xiong S. *et al.* Crosstalk between endoplasmic reticulum stress and oxidative stress: a dynamic duo in multiple myeloma. *Cell Mol Life Sci.* **78**: 3883-3906 (2021)

[2] Caillot M. *et al.* Targeting reactive oxygen species metabolism to induce myeloma cell death. *Cancers (Basel).* **13**:2411(2021)

Comment 20: The authors can demonstrate good rescue efficiency by using GSH. However, GSH has a caveat as it can also be broken down into glycine and therefore (for example) also rescue purine synthesis in parallel. To corroborate that the rescue effect is really related to ROS, alternative ROS scavengers such as Trolox would be a good additional control.

Response to comment 20:

Thank you for your constructive suggestions. Following your suggestion, MM cell lines were treated with a combination of BTZ and Trolox. As shown in Figure R18, Trolox rescues the inhibition of proliferation caused by BTZ, suggesting that the rescue effect is likely related to ROS (Fig. R18a became Fig. 5c, new data; Fig. R18b-c became Extended Data Fig. 6d-e).

Figure R18. Glycine deprivation inhibit cell proliferation by increasing ROS levels in MM cells.

a ARP1 cells cultured with (10 mg/L) or without exogenous glycine were treated with different doses of BTZ (0, 2.5, 5, 10, or 20 nM) and with or without Trolox (100 µM) for 48 h, followed by a CCK-8 assay (mean ± sd, n = 3 in each group). **b** CCK-8 assay in ARP1 Scramble (Scr) and ARP1 SLC6A9sh1 cells exposed to different doses of BTZ (0, 5, or 10 nM) and with or without Trolox (100 µM) (mean ± sd, n = 3 in each group). **c** ROS levels in ARP1 Scr and ARP1 SLC6A9sh1 cells treated with BTZ (5 nM) and with or without Trolox (100 µM) (mean ± sd, n = 3 in

each group). * $p < 0.05$, *** $p < 0.001$, ns: not significant ($p > 0.05$). p values were calculated using two-tailed unpaired Student's t -tests (a, b, c).

Comment 21: It is not clear to me why MM cells (in contrast to other cells) can not survive on serine alone. Thus, another possible explanation on why MM cells consume glycine could be that such cells have a defect in serine consumption. In this case, an alternative substrate could be glycine and formate to sustain purine and GSH synthesis. Did the authors check if the cells are able to consume serine? Related to this possibility: do the cells release or consume formate?

Response to comment 21:

Thank you very much for your question and critical comments. A set of non-targeted metabolomics data for MM cell lines conditioned media showed that serine was elevated in conditioned media compared with fresh medium (Fig. R19a). However, glycine was reduced in the conditioned media of MM cells (Fig. R19b). Formate was not detected in conditioned media. Most of the serine-derived one-carbon units are released from cells as formate^[1]. However, in the present study, MM cells release serine but not release formate, suggesting low level of serine consumption in MM cells. In addition, MM cells release serine, reducing the metabolism of intracellular serine into glycine. Therefore, exogenous glycine is taken up to meet the need for proliferation of MM cells.

Figure R19. MM cells absorb glycine and release serine.

a The abundance of serine in fresh medium (Control) and conditioned medium derived from five MM cell lines (mean \pm sd, $n = 3$ in each group). **b** The abundance of glycine in fresh medium (Control) and conditioned medium derived from five MM cell lines (mean \pm sd, $n = 3$ in each group). * $p < 0.05$, ** $p < 0.01$, ns: not significant ($p > 0.05$). p values were calculated using two-tailed unpaired Student's t -tests (a, b).

[1] Meiser J. *et al.* Serine one-carbon catabolism with formate overflow. *Sci Adv.* 2: e1601273 (2016)

REVIEWER COMMENTS

Reviewer #1 (Remarks to the Author):

The revised manuscript is much improved and all major concerns have been addressed

Reviewer #2 (Remarks to the Author):

The comments appear to be addressed.

Reviewer #3 (Remarks to the Author):

The authors addressed most of my concerns and significantly improved the quality of the manuscript. Overall, I agree to the main conclusion of this manuscript that “glycine is a key metabolic regulator of MM” and I principally endorse publication of the manuscript. I also agree that glycine plays an important role as a precursor for GSH synthesis but I remain with two technical concerns:

1. The purine rescue experiment presented in figure R17c have the caveat that the used purine concentration of 5 nM is very low. A general approach to rescue purine synthesis defects is by providing hypoxanthine (in the μM range to cells). In my lab we usually use 50 μM (Factor 10.000). Therefore, I am not sure if the drawn conclusion (glycine is not relevant for purine synthesis in MM cells) can be supported by this data.

The reported gH2AX effects reported in the manuscript can also be caused from purine nucleotide depletion which also causes DNA damage. While it is clear that the reported phenotype related to DNA damage associates with glycine availability it is not proven whether it is specifically caused by ROS or nucleotide depletion (likely both aspects contribute).

To me it still appears that glycine is important for both nucleotide synthesis and GSH synthesis. This is also supported by the ^{13}C tracing data.

2. I still have concerns if there is a meaningful rate of glycine catabolism via the glycine cleavage system. In my view it is (still) not supported by the data reported in figure R7a.

My concern relates to the natural isotope abundance correction approach. I am not so sure if this equation correctly accounts for the probability of each isotopologue (beyond M+1). In my view, the larger the molecule the higher the chance of an error. According to current standards in the field, the best approach for the correction of natural isotope abundance is a purely mathematical approach as described in here: PMID: 25731751 (and available as scripts and additional protocols online).

If I understand the explanation by the authors correct, their approach bears the risk that it can trigger artefacts when metabolite levels change over time, because they use the peak areas of M+1 at time point 0. If absolute metabolite levels increase over time, their absolute M+1 peak level will increase in a linear relation. That said, even in full absence of a ^{13}C tracer, the correction approach might reveal

certain M+1 labelling for any metabolite that increases at later time points (compared to time point 0). I am also surprised to see the extremely high M+1 labeling of xanthine. Xanthine is a breakdown product of purines, thus I would expect a similar labelling pattern as AMP, GMP and IMP. How do the authors explain the huge M+1 label in xanthine? How do the GSH and xanthine levels behave between time point 0 and 18?

Independent of these technical concerns, even when all the corrections are correct and there is indeed some M+1 label in GSH derived from glycine. The fractional contribution of GCS to GSH production compared to glycine as a direct substrate (M+2) is very little.

On the other hand, the functional data on GLDC in figure 4 is convincing and suggests that GLDC plays a role. However, as written above, the ^{13}C tracing does not clearly support this. Maybe GLDC has additional (moonlighting) functions, that go beyond glycine catabolism.

I do not request any additional experiment, but I would consider rephrasing the text a bit.

Finally, I would like to reiterate that I endorse publication in principle, when the technical concern has been addressed.

Dear reviewers:

Thanks for your comments concerning our manuscript entitled "**Inhibition of glycine metabolism suppresses the progression and drug resistance of multiple myeloma** " (Full Paper, ID NCOMMS-21-05532A-Z). Those comments are all valuable and very helpful for us, both for revising and improving our paper and for guiding our research. We have studied the comments carefully and have made corrections which we hope will meet the publication standard. The revised portions are marked in red in the paper. The main corrections in the paper and the responses to the reviewers' comments are as follows:

Reviewer #3 (Remarks to the Author):

The authors addressed most of my concerns and significantly improved the quality of the manuscript.

Overall, I agree to the main conclusion of this manuscript that “glycine is a key metabolic regulator of MM” and I principally endorse publication of the manuscript. I also agree that glycine plays an important role as a precursor for GSH synthesis but I remain with two technical concerns:

1. The purine rescue experiment presented in figure R17c have the caveat that the used purine concentration of 5 nM is very low. A general approach to rescue purine synthesis defects is by providing hypoxanthine (in the μM range to cells). In my lab we usually use 50 μM (Factor 10.000). Therefore, I am not sure if the drawn conclusion (glycine is not relevant for purine synthesis in MM cells) can be supported by this data.

The reported $\gamma\text{-H}_2\text{AX}$ effects reported in the manuscript can also be caused from purine nucleotide depletion which also causes DNA damage. While it is clear that the reported phenotype related to DNA damage associates with glycine availability it is not proven whether it is specifically caused by ROS or nucleotide depletion (likely both aspects contribute).

To me it still appears that glycine is important for both nucleotide synthesis and GSH synthesis. This is also supported by the ^{13}C tracing data.

Response to comment 1: We greatly appreciate your valuable comments. We used

hypoxanthine (50 μ M) and GSH (10 μ M) to treat ARP1 cells cultured with glycine-free medium in the revised experiment based on your suggestion. As a result, additions of GSH or hypoxanthine were effective in rescue cell proliferation in ARP1 cells cultured with glycine-free medium (Figure R1a), indicating that exogenous glycine contributes to cell proliferation through promoting synthesis of both GSH and purines in MM cells. These data supported your opinion that glycine is important for both nucleotide synthesis and GSH synthesis. These results are consistent with the previous publication that exogenous glycine contributed to cell proliferation through promoting purines synthesis in melanoma cells [1]. These results were included in Extended data Fig. 3 in our revised manuscript.

To determine whether glycine deprivation induced DNA damage is caused by ROS or nucleotide depletion, ARP1 cells cultured with glycine-free medium were treated with glycine (10 mg/L), GSH (10 μ M), hypoxanthine (50 μ M) and Trolox (100 μ M) for 48 hours and followed by detection of γ -H₂AX by using Western blot. As shown in Figure R1b, additions of GSH, hypoxanthine and Trolox could decrease the expression of γ -H₂AX in ARP1 cells cultured with glycine-free medium. Trolox is an analogue of vitamin E with a powerful antioxidant effect and effective in reducing ROS-induced DNA damage [2]. In view with these data, we think that both ROS and nucleotide depletion contribute to glycine deprivation-induced DNA damage.

In the revised manuscript, Figure R1a was added in Extended data Fig. 3d. The describe of new figure was highlight in red in the first paragraph of Result 3 in the revised manuscript.

Figure R1. GSH and hypoxanthine contribute to proliferation and inhibit DNA damage of MM cells.

a CCK-8 assay of ARP1 with or without GSH and hypoxanthine. **b** Western blots of γ -H₂AX and β -actin in ARP1

cells cultured with or without glycine, GSH (10 μ M), hypoxanthine (50 μ M) and Trolox (100 μ M) for 48 h.

[1] Jain M, *et al.* Metabolite profiling identifies a key role for glycine in rapid cancer cell proliferation. *Science*. 336, 1040-1044 (2012).

[2] Forrest VJ, *et al.* Oxidative stress-induced apoptosis prevented by Trolox. *Free Radic Biol Med*. 16, 675-84 (1994).

2. I still have concerns if there is a meaningful rate of glycine catabolism via the glycine cleavage system. In my view it is (still) not supported by the data reported in figure R7a.

My concern relates to the natural isotope abundance correction approach. I am not so sure if this equation correctly accounts for the probability of each isotopologue (beyond M+1). In my view, the larger the molecule the higher the chance of an error. According to current standards in the field, the best approach for the correction of natural isotope abundance is a purely mathematical approach as described in here: PMID: 25731751 (and available as scripts and additional protocols online).

If I understand the explanation by the authors correct, their approach bears the risk that it can trigger artefacts when metabolite levels change over time, because they use the peak areas of M+1 at time point 0. If absolute metabolite levels increase over time, their absolute M+1 peak level will increase in a linear relation. That said, even in full absence of a ^{13}C tracer, the correction approach might reveal certain M+1 labelling for any metabolite that increases at later time points (compared to time point 0). I am also surprised to see the extremely high M+1 labeling of xanthine. Xanthine is a break down product of purines, thus I would expect a similar labelling pattern as AMP, GMP and IMP. How do the authors explain the huge M+1 label in xanthine? How do the GSH and xanthine levels behave between time point 0 and 18?

Independent of these technical concerns, even when all the corrections are correct and there is indeed some M+1 label in GSH derived from glycine. The fractional contribution of GCS to GSH production compared to glycine as a direct substrate (M+2) is very little.

On the other hand, the functional data on GLDC in figure 4 is convincing and suggests that GLDC plays a role. However, as written above, the ^{13}C tracing does not

clearly support this. Maybe GLDC has additional (moonlighting) functions, that go beyond glycine catabolism.

I do not request any additional experiment, but I would consider rephrasing the text a bit.

Finally, I would like to reiterate that I endorse publication in principle, when the technical concern has been addressed.

Response to comment 2: Thank you very much for pointing the abnormal of xanthine. We carefully checked the original data of metabolic flux experiment and then found that we might mistakenly identified guanine as xanthine. Xanthine is very close to guanine at molecular weight (xanthine: 152.111 Dalton, guanine: 151.126 Dalton). In the revised experiment, we detected both xanthine and guanine by liquid chromatography triple-quadrupole mass spectrometry in the remaining samples derived from previous metabolic flux experiment. As shown in Figure R2a, the retention time of xanthine (green box) is close to the retention time of guanine (blue box). In the original manuscript, we mistakenly identified the peaks of guanine (m+1) as the peaks of xanthine (m+0), identified the peaks of guanine (m+2) as the peaks of xanthine (m+1), identified the peaks of guanine (m+3) as the peaks of xanthine (m+2). We are very sorry for our mistake. In the revised manuscript, we added the data of guanine in the revised Figure 3a-b, Figure S3a, Figure S3d and the description of guanine were highlight in red in the revised manuscript.

We greatly appreciate your comments about the natural isotope abundance correction approach. We carefully checked the original data of our metabolic flux experiment. In our metabolic flux experiment, we corrected the natural isotope abundance through reference sample which was collected at 0 h based on previous publications^[1, 2]. The reference samples (1×10^7 cells / sample) and experimental samples (1×10^7 cells / sample) were detected with the same method at the same time, so we think that the reference samples could reflect the real natural isotope abundance in our cells. To correct the natural isotope abundance, the proportion (P_0) of natural ^{13}C -labeled metabolite relative to non ^{13}C -labeled metabolite in ARP1 cells collected at 0 h

was obtained. $P_0 = \text{abundance of natural } ^{13}\text{C-labeled metabolite}_{0h} / \text{abundance of non } ^{13}\text{C labeled metabolite}_{0h}$. The abundance of natural ^{13}C -labeled metabolite in ARP1 cells cultured with $^{13}\text{C}_2$ -glycine was calculated by following formula: abundance of natural ^{13}C -labeled metabolite = abundance of non ^{13}C -labeled metabolite $\times P_0$. Then the abundance of natural ^{13}C -labeled metabolite was subtracted from the ^{13}C -labeled metabolite in ARP1 cells cultured with $^{13}\text{C}_2$ -glycine. The MDVs of ^{13}C -labeled metabolites were obtained by following formula: MDVs = (abundances of ^{13}C -labeled metabolites - abundances of natural ^{13}C -labeled metabolites) / (abundances of total metabolites - abundances of natural ^{13}C -labeled metabolites).

As shown in Figure R2B, the absolute levels of total GSH, GSH (m+1) and GSH (m+2) are increased over time. The absolute level of natural ^{13}C -labeled GSH may increase with the increase of total GSH, however, the proportion (P) of natural ^{13}C -labeled GSH relative to non ^{13}C -labeled GSH will not change over time. In our study, the abundance of natural ^{13}C -labeled GSH in ARP1 cells cultured with $^{13}\text{C}_2$ -glycine was calculated through following formula: abundance of natural ^{13}C -labeled GSH = abundance of non ^{13}C -labeled GSH $\times P$. The MDVs of GSH in ARP1 cells cultured with $^{13}\text{C}_2$ -glycine were calculated after subtracting natural ^{13}C -labeled GSH from total ^{13}C -labeled GSH. As a result, the MDVs of GSH (m+2) are increased over time, and the MDVs of GSH (m+1) are changed slightly and above 0.05 (Figure R2c). In view of these data, we considerate that GSH (m+1) is exist in ARP1 cells cultured with $^{13}\text{C}_2$ -glycine.

We greatly appreciate your suggestions about GLDC. We agree with your opinion that GLDC has additional functions besides glycine cleavage metabolism. We referred several publications about GLDC, and found that GLDC is involved in multiple biological processes, including glycine cleavage metabolism^[3], glycolysis^[4-6], purines synthesis^[7], GSH synthesis^[8, 9], autophagy^[10], etc. The proportion of GSH (m+1) in total GSH is much lower than GSH (m+2), indicating that GSH production from GCS (glycine cleavage system) is less than glycine as a direct substrate. GLDC plays an important role in maintaining the balance of glycine in microenvironment with high glycine^[11]. Therefore, our results suggested that glycine cleavage metabolism is exist

in MM cells. In addition, we think that GLDC may contribute to GSH through other mechanisms besides glycine cleavage metabolism. At the end of the fourth result section, we mentioned the other functions of GLDC and concluded that glycine cleavage metabolism is just one of mechanisms by which GLDC promotes GSH synthesis and cell proliferation in MM cells. We also mentioned other mechanisms by which GLDC promotes GSH synthesis and cell proliferation besides glycine cleavage metabolism at the first paragraph of discussion section. We revised accordingly the results and discussion in red in our revised manuscript.

Again, we greatly appreciate your professional and valuable comments and suggestions. Your suggestions significantly improved our manuscript. We look forward to a favorable decision from you.

Figure R2 Analysis of metabolic flux. a The retention time of xanthine and guanine. **b** The absolute levels of total GSH, GSH (m+1) and GSH (m+2) in ARP1 cells collected at 0 h, 2 h, 6 h, and 18 h. **c** the MDVs of GSH (m+1) and GSH (m+2) in ARP1 cells collected at 2 h, 6 h, and 18 h.

[1] Trefely S, Ashwell P, Snyder NW. FluxFix: automatic isotopologue normalization for metabolic tracer analysis. *BMC Bioinformatics*. 25; 17 (1): 485 (2016)

[2] Bononi A, *et al.* Germline BAP1 mutations induce a Warburg effect. *Cell Death Differ*. 24 (10): 1694-1704 (2017)

- [3] Go MK, *et al.* Glycine decarboxylase is an unusual amino acid decarboxylase involved in tumorigenesis. *Biochemistry*. 53 (5): 947-56 (2014)
- [4] Woo CC, *et al.* Inhibiting Glycine Decarboxylase Suppresses Pyruvate-to-Lactate Metabolism in Lung Cancer Cells. *Front Oncol*. 8: 196 (2018)
- [5] Zhang WC, *et al.* Glycine decarboxylase activity drives non-small cell lung cancer tumor-initiating cells and tumorigenesis. *Cell*. 148 (1-2): 259-72 (2012)
- [6] Jung Y, *et al.* Metabolic signature genes associated with susceptibility to pyruvate kinase, muscle type 2 gene ablation in cancer cells. *Mol Cells*. 35 (4): 335-41 (2013)
- [7] Alptekin A, *et al.* Glycine decarboxylase is a transcriptional target of MYCN required for neuroblastoma cell proliferation and tumorigenicity. *Oncogene*. 38 (50):7 504-7520 (2019)
- [8] Jog R, Chen G, Wang J, Leff T. Hormonal regulation of glycine decarboxylase and its relationship to oxidative stress. *Physiol Rep*. 9 (15): e14991 (2021)
- [9] Zhuang H, *et al.* Downregulation of glycine decarboxylase enhanced cofilin-mediated migration in hepatocellular carcinoma cells. *Free Radic Biol Med*. 120:1-12 (2018)
- [10] Zhuang H, *et al.* Glycine decarboxylase induces autophagy and is downregulated by miRNA-30d-5p in hepatocellular carcinoma. *Cell Death Dis*. 10 (3): 192 (2019)
- [11] Kim D, *et al.* SHMT2 drives glioma cell survival in ischaemia but imposes a dependence on glycine clearance. *Nature*. 520 (7547): 363-367 (2015)

REVIEWERS' COMMENTS

Reviewer #3 (Remarks to the Author):

I would like to thank the authors for carefully addressing my comments, which now are properly addressed and I endorse publication of this manuscript.